# Climate policy implications of nonlinear decline of Arctic land permafrost and other cryosphere elements

Dmitry Yumashev [1,2], Chris Hope[3], Kevin Schaefer [4], Kathrin Riemann-Campe [5], Fernando Iglesias-Suarez[2,6], Elchin Jafarov[4,7], Eleanor J. Burke [8], Paul J. Young [1,2,9], Yasin Elshorbany[10] & Gail Whiteman[1]

Arctic feedbacks accelerate climate change through carbon releases from thawing permafrost and higher solar absorption from reductions in the surface albedo, following loss of sea ice and land snow. Here, we include dynamic emulators of complex physical models in the integrated assessment model PAGE-ICE to explore nonlinear transitions in the Arctic feedbacks and their subsequent impacts on the global climate and economy under the Paris Agreement scenarios. The permafrost feedback is increasingly positive in warmer climates, while the albedo feedback weakens as the ice and snow melt. Combined, these two factors lead to significant increases in the mean discounted economic effect of climate change: $+4.0\%$ ($24.8 trillion) under the 1.5 °C scenario, $+5.5\%$ ($33.8 trillion) under the 2 °C scenario, and $+4.8\%$ ($66.9 trillion) under mitigation levels consistent with the current national pledges. Considering the nonlinear Arctic feedbacks makes the 1.5 °C target marginally more economically attractive than the 2 °C target, although both are statistically equivalent.

[1] Pentland Centre for Sustainability in Business, Lancaster University, Lancaster LA1 4YX, UK. [2] Lancaster Environment Centre, Lancaster University, Lancaster LA1 4YQ, UK. [3] Judge Business School, University of Cambridge, Cambridge CB2 1AG, UK. [4] National Snow and Ice Data Centre, Cooperative Institute for Research in Environmental Sciences, University of Colorado, Boulder CO 80309-0449 CO, USA. [5] Alfred-Wegener-Institut, Helmholtz Zentrum für Polar- und Meeresforschung, Bremerhaven 27515, Germany. [6] Department of Atmospheric Chemistry and Climate Group, Institute of Physical Chemistry Rocasolano, CSIC, Madrid 28006, Spain. [7] Computational Earth Science, Earth and Environmental Sciences EES-16, Los Alamos National Laboratory, Los Alamos, NM, USA. [8] UK Met Office, Exeter EX1 3PB, UK. [9] Data Science Institute, Lancaster University, Lancaster LA1 4YW, UK. [10] College of Arts & Sciences, University of South Florida, St. Petersburg, FL 33701, USA. Correspondence and requests for materials should be addressed to D.Y. (email: d.yumashev@lancaster.ac.uk)

The Arctic region is warming twice as fast as the global average[1], manifested by a decrease in sea ice, snow and glaciers and permafrost degradation relative to their benchmark average states for the period between 1979 and 2005[2–6]. These changes can accelerate global warming further through a variety of climatic feedbacks. Carbon from thawing permafrost released into the atmosphere results in the permafrost carbon feedback (PCF)[7,8]. Decreasing sea ice and land snow covers increase solar absorption in high latitudes, causing the surface albedo feedback (SAF)[9,10]. Both feedbacks amplify the anthropogenic signal.

The PCF and SAF represent three of the thirteen main tipping elements the Earth's climate system identified in recent surveys[11–13]. Tipping elements are physical processes acting as positive nonlinear climate and biosphere feedbacks that, after passing a threshold, could irreversibly shift the planetary system to a new warmer state[13]. They could cause additional impacts on ecosystems, economies and societies throughout the world. The risk of triggering the tipping elements is one of the arguments for adopting the ambitious 1.5 °C and 2 °C targets in the Paris Agreement[14–16]. Therefore, a rigorous quantitative assessment of the climate tipping elements under different climatic and socio-economic scenarios is required to estimate their impacts and narrow down the uncertainties.

Despite significant advances documented by the IPCC 5th Assessment Report (AR5)[6], projections of future climate using general circulation models (GCMs) from the 5th climate model inter-comparison project (CMIP5) do not include the PCF[17,18], although several models are set to incorporate the PCF in their next versions as part of CMIP6. Consequently, most climate policy assessments based on results from the GCMs underestimate the extent of global warming in response to anthropogenic emissions. The SAF, on the other hand, is present in GCM climate projections through the coupling of sea ice and land surface models to atmosphere and ocean models[17]. However, existing estimates of the total economic impact of climate change under different policy assumptions using integrated assessment models (IAMs) assume that radiative forcing from the SAF increases linearly with global mean temperature[19,20], which is inconsistent with predictions of the GCMs[21].

In this paper, we explore nonlinear transitions in the state-dependent PCF and SAF, and estimate the resulting climatic and economic impacts globally. To perform the analysis, we develop dynamic model emulators of the nonlinear PCF and SAF, which are comparatively simple statistical surrogates of the highly complex physical models. The emulators are integrated within PAGE-ICE, a new development of the PAGE09 IAM[19,20] that includes a number of updates to climate science and economics (Methods, Supplementary Note 1). The climatic impacts focus on changes in the global mean surface temperature (GMST) and the economic impacts focus on the net present value (NPV) of the total cost associated with future climate change. We consider a wide range of scenarios: zero emissions after 2020, the 1.5 °C and 2 °C targets for 2100 and the nationally determined contributions (NDCs) from the Paris Agreement, and a business as usual (BaU) scenario. We also introduce an intermediate 2.5 °C target, which requires more mitigation than is proposed by the NDCs, and an NDCs Partial scenario with a persistent under-delivery on pledges consistent with an estimated long-term effect of the US's withdrawal from the Paris Agreement. The scenarios extend out to 2300 to capture the effects of multiple slow physical processes including the PCF and the loss of the winter sea ice under high emissions pathways. While very long horizons like this may appear irrelevant from the point of view of the actual socio-economic processes, the well-established technological, demographic and resource constraints[22,23] imply that the range of scenarios is still plausible beyond the 21st century[24].

In addition to the PCF and SAF, Arctic feedbacks include carbon emissions from thawing sub-sea permafrost, boreal forest uptake and changes in ocean circulation from the melting of the Greenland ice sheet[13,25], which we do not explicitly simulate it in this study. Emissions from thawing sub-sea permafrost on Arctic shelf are poorly understood in comparison with land permafrost emissions[26]. The boreal forest and Greenland ice sheet feedbacks are beyond the scope of this study, along with the non-Arctic tipping elements and other major uncertain elements in the climate system such as the cloud feedback[27]. While not modelled directly, many of these effects are included implicitly in the PAGE-ICE IAM through a number of uncertain climate system parameters constrained according to the latest literature (Methods, Supplementary Note 1).

Our results show that the PCF gets progressively stronger in warmer climates, while the SAF weakens. Both feedbacks are characterised by nonlinear equilibrium responses to warming. The PCF also develops state-dependent lagged behaviour. Compared with zero PCF and constant SAF, which are the legacy values that have been used in climate policy modelling to date, the combined nonlinear PCF and SAF cause statistically significant extra warming globally under the low and medium emissions scenarios. For high emissions scenarios, the strength of the PCF saturates, and the weakening SAF gradually cancels the warming effect of the PCF; for BaU, this takes place from the second half of the 22nd century onwards. Nevertheless, under all scenarios, the predominantly warmer future climate associated with the nonlinear PCF and SAF relative to their legacy values translates into marginal increases in the total discounted economic effect of climate change. These increases, which are significant for all scenarios except for BaU, occur through additional temperature-driven impacts on economy, ecosystems and human health, additional impacts from sea level rise, as well as highly uncertain extra impacts from social discontinuities and climate tipping elements other than the PCF and SAF. Even with the legacy PCF and SAF, emissions pathways in the range between the 1.5 °C and 2 °C targets lead to the lowest total economic effects of climate change compared to all other scenarios. Considering the nonlinear PCF and SAF makes the pathways towards the lower end of the range covered by the Paris Agreement targets marginally more economically attractive.

## Results

**Nonlinear PCF and SAF**. We base the PCF emulator on simulated emissions of thawed permafrost carbon from the two permafrost-enabled global land surface models (LSMs): SiBCASA (Simple Biosphere/Carnegie-Ames-Stanford Approach) and JULES (Joint UK Land Environment Simulator)[7,28,29] (Fig. 1). Permafrost carbon is organic matter buried and frozen in permafrost. The two LSMs have markedly different responses to future climate change: SiBCASA appears to be on the upper end and JULES on the lower end of the reference multi-models studies[8,29,30] (Supplementary Note 2, Supplementary Fig. 1). Their combined use here provides a suitable estimate of the range of permafrost responses arising from uncertainty in LSM parameterizations.

Our uncertainty estimate also depends on the range of global climate model (GCM) outputs used to force the LSMs, accounting for both the structural uncertainty arising from the different GCM's climate sensitivities, as well as the irreducible uncertainty arising from weather and climate variability. To capture this uncertainty in the PCF, both LSMs were forced with output from a range of GCMs, sampling the full range of expected Arctic

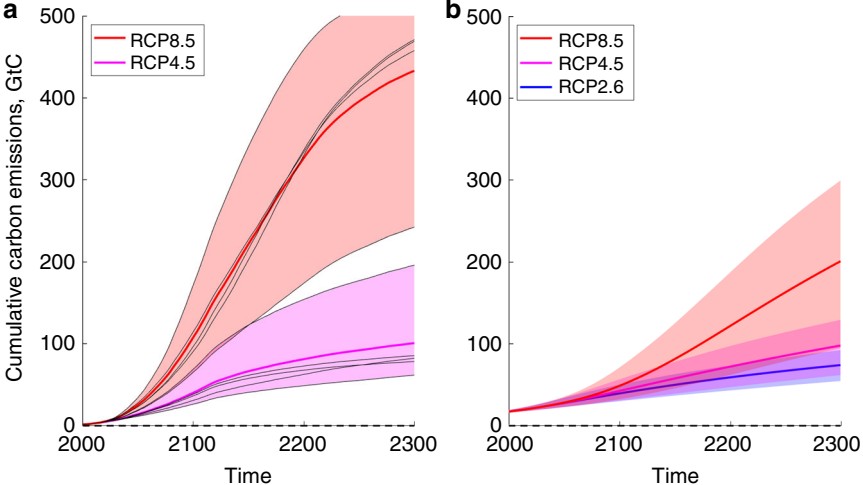

**Fig. 1** Cumulative carbon emissions from thawing land permafrost simulated using specialised land surface models. $CO_2$ component of cumulative emissions of carbon from thawing land permafrost, obtained from **a** SiBCASA and **b** JULES LSMs forced by multiple climate models (GCMs) under a range of climate scenarios out to 2300. Thick lines: multi-GCM means; shaded areas: multi-GCM spread between the lowest and the highest values; thin black lines: SiBCASA runs with individual GCMs. Horizontal black dashed lines: legacy zero permafrost emissions currently assumed in IAMs. Units: GtC. Source data are provided as a Source Data file

responses under a given climate scenario to 2300. SiBCASA is an explicit LSM and was run to 2300 with output from five CMIP5 GCMs under two scenarios, whereas JULES was configured to run to 2300 with output from 22 CMIP3 GCMs under three climate scenarios (Fig. 1). Having information to 2300 is important to capture the nonlinear transitions in the permafrost emissions.

The dynamic PCF emulator uses a statistical fit to SiBCASA and JULES outputs for the land permafrost carbon emissions in the form of $CO_2$ and methane, capturing nonlinear effects seen in the LSM simulations. The PCF emulator only models the emitted permafrost carbon explicitly, while also accounting for the time lags between the temperature rise, thawed carbon and emitted carbon, as well as the uncertainty in the initial permafrost carbon stock[31]. The PAGE-ICE model adds the permafrost fluxes from the PCF emulator to the anthropogenic global $CO_2$ and methane emissions that follow a given climate scenario.

We base the SAF emulator on the ALL/CLR method of calculating the SAF using downward and upward atmospheric transmissivity and reflectivity inferred from climate models (GCMs)[32,33]. The method involves an atmospheric reflectivity parameterisation, which represents the effect of clouds and is based on clear sky and all sky shortwave fluxes diagnosed from the GCMs. It allows us to account for the localised changes to the cloud cover and its effect on the SAF in line with the physical interactions represented in the fully-coupled CMIP5 models (Supplementary Figure 2)[34–36]. We do not compute the global cloud feedback; instead, it is included implicitly in the ECS parameter in the PAGE-ICE model and is assumed to be state-independent (see the section on the robustness of the results below).

We use historic and RCP8.5 simulations of 16 CMIP5 GCMs that have the diagnostic variables required for the SAF calculation (Supplementary Table 1)[32]. While short of the complete CMIP5 ensemble, these models sample the full range of Arctic responses as seen in the whole ensemble. Eight of the models have simulations that extend out to 2300, which is necessary to capture the nonlinear transitions in the SAF. Each model has its own domains for Arctic sea ice and land snow covers based on their respective monthly maximum extents during the pre-industrial period. The SAF, therefore, is separated into the northern hemispheric sea ice, northern hemispheric land snow and rest of

the world. The components are represented as functions of the GMST rise individually for each model, at which point the multi-model statistics is established (Fig. 2). The sea ice and land snow components of the SAF peak for the GMST anomalies between 0–1 °C and 1–3 °C, respectively, coinciding with the loss of the summer sea ice[2,37] and spring and summer land snow[38] covers coupled with high Arctic insolation. However, both components decrease for higher GMST as the sea ice and land snow covers continue to decline, and eventually approach zero when the covers disappear. The plateau in the sea ice SAF component between 5–7 °C coincides with the loss of the spring sea ice[39,40]. In contrast, the SAF for the rest of the world stays nearly constant until high GMST anomalies.

The SAF emulator models the global SAF as a function of the GMST (Methods), reproducing the nearly monotonic decline driven largely by the Arctic sea ice and land snow components. While IAMs other than PAGE-ICE have implied unrealistic constant SAF (dashed lines in Fig. 2), which is implicitly included in the $2 \times CO_2$ ECS parameter (Methods), the SAF emulator accounts for the difference between the nonlinear SAF and its constant legacy value. It alters the governing equation for the GMST change in PAGE-ICE by adding extra terms to the total anthropogenic radiative forcing (RF) for a given climate scenario (Supplementary Note 3).

**GMST changes due to the nonlinear PCF and SAF**. Figure 3 shows the medians and 25–75% ranges for the GMST projections relative to pre-industrial levels for the climate scenarios considered, obtained using PAGE-ICE with the legacy values of the PCF and SAF. For the PCF, the legacy value is zero emissions from permafrost since the PCF is not included in most climate projections using GCMs[18]. For the SAF, the legacy value is constant SAF of $0.35 \pm 0.05$ W m$^{-2}$ K$^{-1}$, which corresponds to $2 \times CO_2$ equilibrium climate sensitivity (ECS) calibrated according to IPCC AR5. In subsequent sections, our main results report the difference between the climatic and economic impacts of the nonlinear PCF and SAF and their respective legacy values. Further details appear in Methods.

Figure 4 shows the means and ±1SD ranges of the absolute changes in GMST until 2300 due to the nonlinear PCF, SAF and PCF & SAF combined, measured relative to their respective

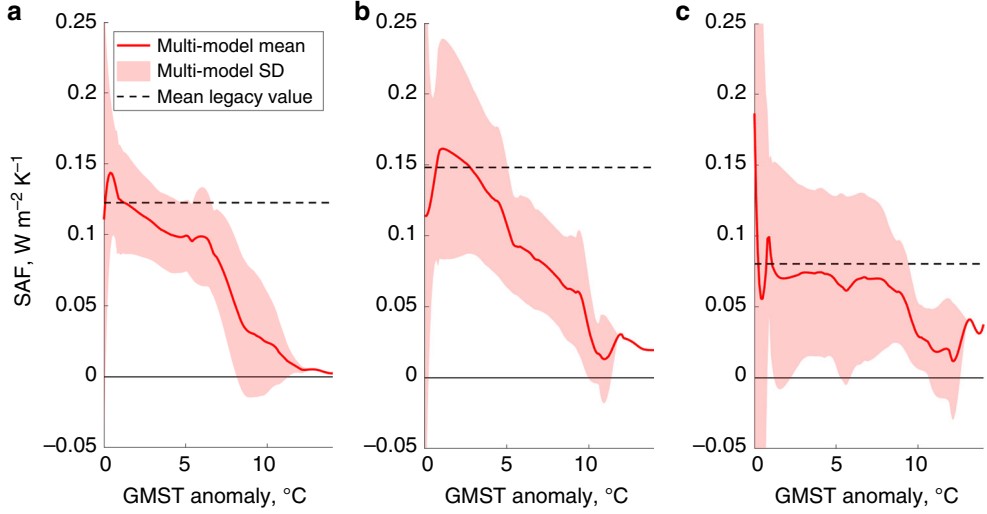

**Fig. 2** Three components of the surface albedo feedback deduced from simulations of fully coupled climate models. SAF components (average global equivalent values) for **a** Arctic sea ice, **b** Northern Hemisphere land snow and **c** rest of the world, presented as functions of the GMST rise relative to pre-industrial conditions. Obtained from multiple CMIP5 GCMs using the ALL/CLR method. Solid red line: multi-model mean; shaded area: ±1 standard deviation (SD). Horizontal black dashed lines: legacy SAF forcing currently assumed in IAMs. Source data are provided as a Source Data file

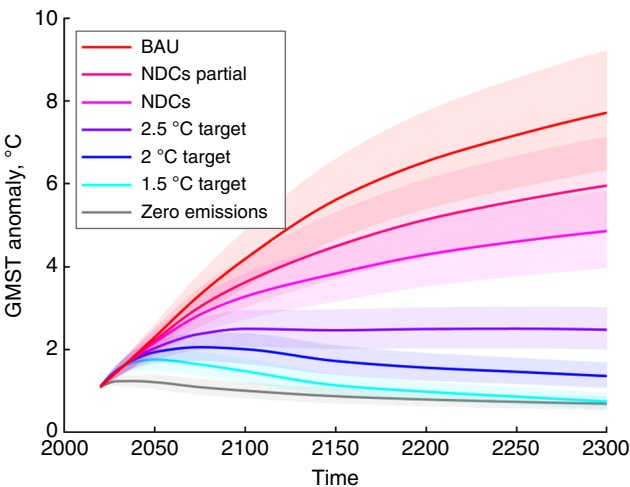

**Fig. 3** Global mean temperature simulations under the range of climate scenarios considered. Median GMST projections relative to pre-industrial 1850–1900 levels (thick lines) and the relevant 25–75% ranges (shaded areas) obtained from 100,000 runs of PAGE-ICE for all the climate scenarios considered, assuming the following legacy values of the PCF and SAF: zero permafrost emissions, constant SAF of 0.35 ± 0.05 W m$^{-2}$ K$^{-1}$. This serves as a base estimate for the subsequent analysis of the nonlinear PCF and SAF. Source data are provided as a Source Data file

legacy values. The GMST changes from the PCF and SAF are smaller than the underlying uncertainty in the base climate projections in PAGE-ICE with legacy constant SAF and zero PCF (Fig. 3; note the different vertical scale). However, with few exceptions, the values plotted in Fig. 4 represent statistically significant shifts in the state of the climate system due to the two feedbacks at the 95% confidence level (the exceptions are listed in the Fig. 4 caption).

Because the legacy value is zero, the PCF increases GMST for all scenarios (Fig. 4a). The slow response of permafrost to thaw means the change in GMST before 2100 due to the PCF is nearly the same for all scenarios except Zero Emissions. The difference between the scenarios only becomes apparent in the 22nd and 23rd centuries, with the GMST effect of the PCF becoming

progressively stronger as emissions increase towards BaU. The GMST increases are virtually indistinguishable between NDCs, NDCs Partial and BaU because the marginal effect on GMST of additional $CO_2$ emissions from the PCF drops as total atmospheric $CO_2$ concentrations increase. In addition, the highest emissions scenarios exhaust the permafrost carbon stocks in some simulations, causing a drop in the annual $CO_2$ flux from permafrost beyond 2200. This results in carbon removal from the atmosphere through $CO_2$ ocean uptake (Supplementary Note 4, Supplementary Fig. 3) and causes a slight decline in the GMST effect of the PCF in the 23rd century for the BaU scenario.

The nonlinear SAF is dominated by the decrease of its sea ice and land snow components (Fig. 2), resulting in less warming and negative GMST changes compared to the constant legacy SAF (Fig. 4b). The NDCs, NDCs Partial and BaU scenarios have the largest temperature increases and greatest decreases in land snow and sea ice SAF components, and thus show the greatest negative differences in GMST. The differences are the highest for BaU with nearly ice-free oceans and snow-free land even in winter after 2200. For the Zero Emissions, 1.5 °C, 2.0 °C and 2.5 °C scenarios, there are small increases in GMST for the entire time period due to the small peaks in the sea ice and land snow SAF components within this temperature range (Fig. 2). Overall, the constant legacy SAF appears reasonable for low emission scenarios, but overestimates GMST for high emission scenarios according to the current generation of climate models (CMIP5), with no apparent tipping points (Supplementary Fig. 4).

The nonlinear SAF can partially compensate for the PCF (Fig. 4c). For the high emission NDC, NDC Partial, and BaU scenarios, reduced warming due to the nonlinear response of the SAF partially cancels out warming due to the PCF. The effect is strongest for the BaU scenario, where the change in GMST from the legacy value switches sign from positive to negative. The SAF slightly amplifies the PCF for the low emission scenarios where the constant SAF forcing assumption remains valid (Zero Emissions, 1.5 °C, 2.0 °C and 2.5 °C). This means IAMs that do not include the PCF and assume a constant SAF will underestimate GMST by between 0.1 and 0.2 C for all but the highest emissions scenarios. These findings stemming from the nonlinearities both in the PCF and SAF have been overlooked in climate policy studies so far (Supplementary Note 5)[41–44].

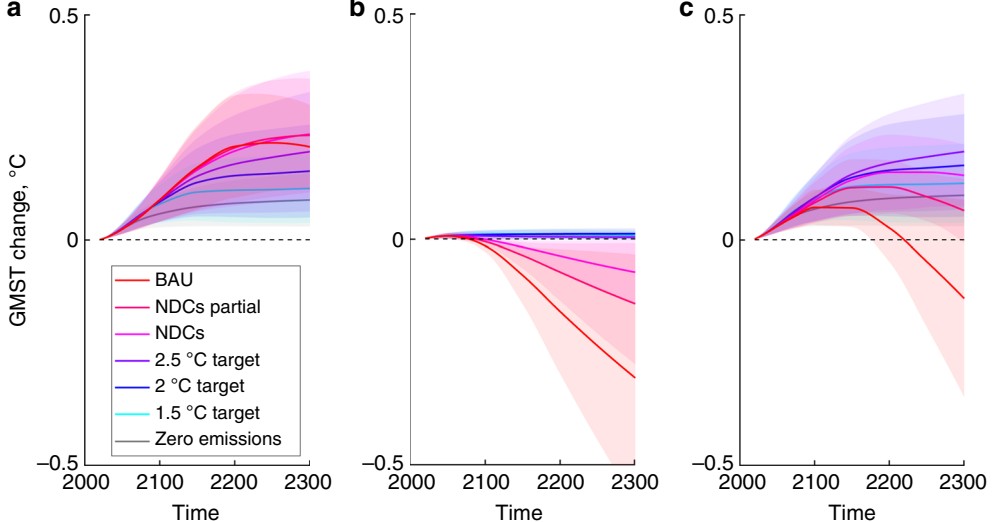

**Fig. 4** Additional warming due to the nonlinear Arctic feedbacks relative to their legacy values. Differences between the GMST effects of the nonlinear PCF and SAF, and the GMST effects of their constant legacy values, obtained using the new PCF and SAF emulators in PAGE-ICE for **a** PCF, **b** SAF and **c** combined PCF & SAF. The legacy PCF value is zero and the legacy SAF value is the black dashed line in Fig. 2. Thick lines: ensemble mean; shaded areas: ±1SD. The cases when the GMST effect of the feedbacks is not significant at the 5% level: SAF under NDCs and NDCs Partial at the turn of the 22nd century, SAF under BaU in the second half of the 21st century, PCF & SAF combined under NDCs Partial in the second half of the 23rd century, and PCF & SAF combined under BaU from the second half of the 22nd century onwards. Source data are provided as a Source Data file

**Implications for the total economic effect of climate change.** The NPV of the total economic effect of climate change, denoted as $C_{NPV}$, consists of mitigation costs, adaptation costs and climate-related economic impacts aggregated until 2300 and discounted using equity weighting and a pure time preference rate[45]. We base the economic impacts due to changing temperatures on a recent macro-econometric analysis of historic temperature shocks on economic growth in multiple countries[46]. We project the economic impact function derived from this analysis onto the 8 global regions of the PAGE model[19] using gridded population-weighted ERA-Interim reanalysis data[47] for mean climatological temperatures in the base year, and adapt it to fit with the consumption-only approach for climate impacts in PAGE with no lasting effects on economic growth. Termed the level effects[46], this provides a likely lower end estimate for the economic impacts and also allows one to compare directly with the default PAGE09 impact functions[48–51], for which the original results for the PCF were derived[41]. We also carry out updates to the sea level rise driver, discontinuity impacts and mitigation costs according to the latest literature (Methods, Supplementary Note 1). The Zero Emissions scenario provides a hypothetical upper bound for the mitigation costs and includes residual impacts from historic emissions.

First, we calculated $C_{NPV}$ for the global climate-economy system using the base PAGE-ICE model with the legacy Arctic feedbacks, and PAGE-ICE with the nonlinear PCF and SAF representations (Fig. 5a). In both settings, the mean total economic effects of the 1.5 °C, 2 °C and 2.5 °C scenarios are the lowest of the seven scenarios considered, while the NDC scenarios and, particularly, the BaU scenario have much higher mean total economic effects. All the distributions have long upper tails representing a possibility of large impacts relative to the means. The tails get elongated for higher emissions scenarios and when the nonlinear PCF and SAF representations are used.

We then calculated the additional economic effect of the nonlinear PCF and SAF relative to the legacy values (Fig. 5b). The nonlinear PCF leads to statistically significant increases in $C_{NPV}$ at the 5% significance level for all the scenarios considered, especially the NDC and BaU. The nonlinear correction to the SAF

leads to small but statistically significant increases in $C_{NPV}$ for Zero Emissions and 1.5 °C, 2.0 °C and 2.5 °C target scenarios, statistically significant decreases in $C_{NPV}$ for NDCs Partial and BaU, and is not significant for NDCs (all at the 5% level).

When the nonlinear PCF and SAF representations are combined, the statistical mean of the economic effect of climate change increases relative to the base estimate with the legacy PCF and SAF by $16.1 trillion (1 trillion = $10^{12}$) for the counterfactual Zero Emissions scenario ($1288 trillion base estimate), $24.8 trillion for 1.5 °C target ($613 trillion base estimate), $33.8 trillion for 2.0 °C target ($613 trillion base estimate), $50.3 trillion for 2.5 °C target ($815 trillion base estimate), $66.9 trillion for NDCs ($1390 trillion base estimate), and by $59.8 trillion for NDCs Partial ($1702 trillion base estimate). These increases are statistically significant (5% level). We also found marginal but statistically insignificant increases for BaU ($2197 trillion base estimate), which remains the most expensive and least desirable scenario.

The mean economic impact of net additional warming from the nonlinear PCF & SAF peaks at just under $70 trillion (NPV until 2300) for NDCs. To put this number into context, it exceeds the estimated long-term gains from economic development in the Arctic region through transit shipping routes[52] and mineral resource extraction[53] under high emissions scenarios by around 10 times, and could also dwarf pan-Arctic damages to infrastructure from thawing permafrost[54,55]. The economic losses due to climate warming also tend to be higher in warmer poorer regions such as India and Africa[20], which are also less likely to benefit from the economic opportunities associated with a warmer Arctic[56].

**Robustness of the estimates for the PCF and SAF effects.** Other major feedbacks implemented in the fully-coupled GCMs such as clouds, water vapour and lapse rate contribute to overall uncertainty and state-dependency in the ECS parameter[35]. The combined magnitude of these feedbacks showed weak responses to GMST increases in CMIP3 GCMs[21]. CMIP5 GCMs, however, produced increases in the water vapour feedback in warmer climates associated with rising tropopause[57]. While the

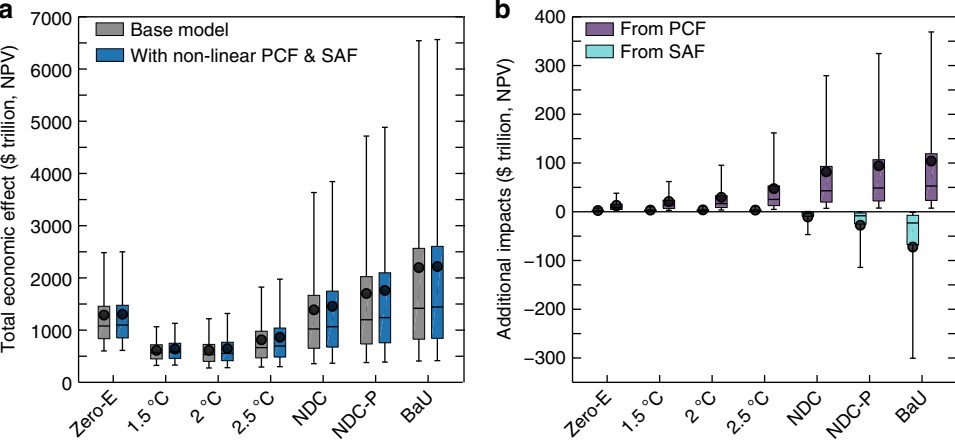

**Fig. 5** Total economic effect of climate change with the nonlinear and legacy Arctic feedbacks. **a** NPVs of the total economic effect of climate change (until 2300, equity-weighted, PTP-discounted), plotted for the legacy Arctic feedbacks (Base model) and with the nonlinear PCF & SAF corrections included, and **b** NPVs of the additional economic impact separately from the nonlinear PCF and SAF, calculated relative to their legacy values under the climate scenarios considered. Whiskers: 5–95% range; boxes: 25–75% range; horizontal lines: median; dots: mean. 100,000 Monte-Carlo runs of PAGE-ICE. The effect of the imaginary Zero Emissions scenario is higher compared to the 1.5 °C, 2 °C and 2.5 °C scenarios due to the large mitigation costs. Source data are provided as a Source Data file

state-dependencies in the planetary feedbacks require further investigations as part of CMIP6, the evidence so far suggests that apart from the SAF effects presented here, the magnitudes of the feedbacks are less likely to decrease with GMST. This implies that our estimates for the impacts of the state-dependent PCF and SAF are likely to be on a conservative side.

The particularly large uncertainty in climate warming caused globally by clouds and aerosol parametrization is an established issue[58–61]. Of the two most recent studies on the cloud feedback that were based on observational constraints, one matched closely with the ECS parameterisation from IPCC AR5 adopted in PAGE-ICE, suggesting that our climate projections are robust. The uncertainties in the permafrost models used in this study, robustness of our PCF and SAF emulators, and uncertainties in other key parameterisations such as the carbon cycle, sea level rise, mitigation business as usual pathway and economic impacts of rising temperatures are discussed in Methods and Supplementary Notes 1–3.

## Discussion

We have investigated the climatic and economic impacts of two major planetary feedbacks associated with the decline of Arctic land permafrost, snow and sea ice. PCF is caused by additional $CO_2$ and methane emissions from thawing permafrost, and SAF is mostly driven by increased solar absorption due to the decline of Arctic sea ice and land snow covers. These two feedbacks belong to the main tipping elements in the Earth's climate system identified by recent surveys[13]. Model simulations indicate that both feedbacks accelerate the warming and are nonlinear, with the PCF being the stronger of the two, while most climate policy studies to date have assumed constant positive SAF and zero PCF, which we refer to as the legacy values. All this warrants their rigorous quantitative assessment. To perform such an assessment, we developed novel statistical emulators of the two Arctic feedbacks calibrated according to simulations results from the specialised land surface and general circulation models. The emulators allow one to study the entire parameter space, which is not possible with complex physical models, and also help establish dynamic links between highly specialized climate and economic models. We implemented the emulators dynamically

inside the new integrated assessment model (IAM) PAGE-ICE, allowing us to explore nonlinear interactions between the Arctic feedbacks and the global climate and socio-economic systems under a range of scenarios consistent with the Paris Agreement.

With the current parameterisations in PAGE-ICE, adding the significant corrections from the nonlinear Arctic feedbacks to the base estimates of the mean total economic effect of climate change makes the 1.5 °C target ($638 trillion) marginally more economically attractive than the 2 °C target ($646 trillion). While the total economic effects of the 1.5 °C and 2 °C scenarios are statistically equivalent (Fig. 6), we have several reasons to believe it would be prudent to aim for emissions towards the bottom end of the range covered by these scenarios. First, the PAGE-ICE model, in common with other aggregate IAMs, does not explicitly model other known climatic tipping elements such as Amazon rainforest, boreal forest, coral reefs and El Niño–Southern Oscillation (ENSO), as well as ocean acidification and climate-induced large-scale migration and conflict[62] (we cannot reject the null hypothesis that the total economic effects of climate change are the same for these scenarios either at the 5% or at the 10% significance level). Some of these effects are already included implicitly in the highly uncertain non-economic and discontinuity impact sectors in PAGE-ICE, contributing to the long upper tails in the distributions of the total economic effect of climate change in Fig. 5a; even with the current parameterisations, the upper tails in the distributions are at their lowest for the 1.5 °C scenario. It is possible that with an explicit modelling of the other climatic and societal tipping elements, as well as with comprehensive representation of the impacts of rising temperatures and increasing extreme weather events on economic growth[63], both the economic effect of climate change with legacy Arctic feedbacks, and the additional impacts due to the nonlinear PCF and SAF, would be higher compared to those reported here. The associated global risks are minimised at lower emissions. Second, it is possible that recent reduction trends in the costs of mitigation technologies such as solar power[64,65], which are captured by PAGE-ICE, could accelerate further if appropriate policy instruments such as carbon prices are implemented globally. Third, PAGE-ICE does not account for possible co-benefits of deep mitigation as part of a wider green growth transition in

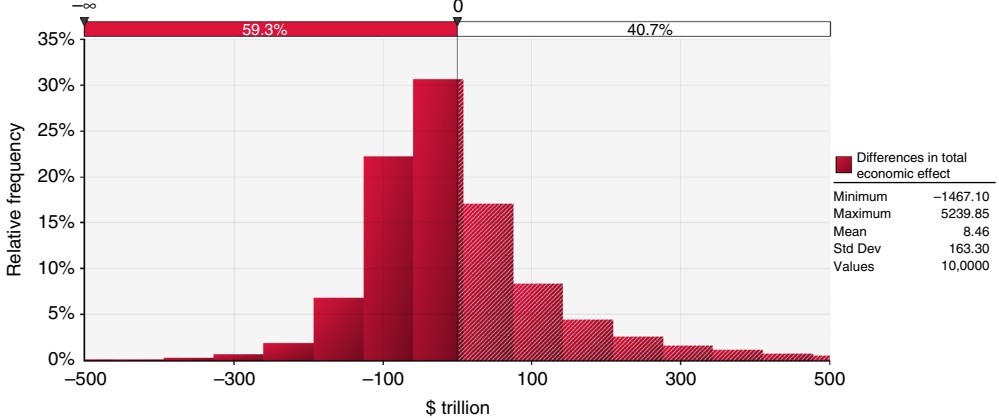

**Fig. 6** Simulated difference between the total economic effects of climate change under the 2 °C and 1.5 °C scenarios. Probability density function (relative frequency) of the difference between the total economic effects of climate change for the 2 °C and 1.5 °C target scenarios with the nonlinear PCF & SAF. Horizontal axis units: $trillion, NPV until 2300, equity-weighted, PTP-discounted. 100,000 Monte-Carlo runs of PAGE-ICE. Source data are provided as a Source Data file

**Table 1 Climate and socio-economic scenarios obtained by pairing RCPs with SSPs**

| Scenario | Description |
|---|---|
| Zero Emissions | GHG emissions stop immediately after 2020 |
| 1.5 °C Target | 50% chance of staying below 1.5 °C relative to pre-industrial in 2100 |
| 2 °C Target | 50% chance of staying below 2 °C relative to pre-industrial in 2100 |
| 2.5 °C Target | 50% chance of staying below 2.5 °C relative to pre-industrial in 2100 |
| NDCs | Current nationally determined contributions (pledges) to reducing GHG emissions |
| NDCs Partial | Around 30% of the NDCs are not met, consistent with long-term effects of the US's withdrawal |
| Business as usual (BaU) | Projections for GHG emissions without NDCs |

economy[66,67]. All these factors advocate for pursuing the target well below 2 °C as the way of avoiding substantial ecological and socio-economic losses from climate change (see Supplementary Discussion for further details)[68].

The nonlinear transitions in the two feedbacks explored in this study demonstrate the pressing need for a better understanding of state-dependent processes in the Earth's climate system, both those associated with the Arctic and beyond. This is important because triggering these and other planetary feedbacks might accelerate the pace of climate change[13,69] and increase the risks of irreversible socio-economic losses[70]. The methodology introduced in this paper could be used to quantitatively assess the economic and climate policy implications of the other tipping elements in the Earth's climate system, including the Greenland and West Antarctic ice sheets, Amazon rainforest, boreal forest, Sahel and ENSO[13]. Such assessments could provide a more complete understanding of the socio-economic risks from climate change that in turn can help guide policymakers towards prudent decisions on emissions reduction targets.

## Methods

**Climate scenarios and model setup in PAGE-ICE.** We defined the scenarios consistent with the Paris Agreement and current climate change projections by pairing representative concentration pathways (RCPs) and shared socio-economic pathways (SSPs) according to the feasible ranges of emissions for each of the five main SSPs[22,23]. Table 1 summarises the scenarios. The imaginary Zero Emissions scenario in which all global emissions stop in the base year 2020 characterises the effect of the historic emissions on the PCF and SAF.

First, we defined a new SSPM scenario by averaging SSP2, SSP3 and SSP4 with equal weights, and paired it with RCP4.5 to represent a likely world with medium levels of emissions. Second, we paired SSP1 with RCP2.6 and SSP5 with RCP8.5, which represents the likely lower and the upper ends of the emissions range and the associated socio-economic makeup of the world. Using these low, medium and

high emissions pairs, we introduced a weighting scheme that covers the entire range as the weighting parameter $w$ changes from $-1$ (lower end) to $+1$ (upper end):

$$\begin{Bmatrix} \text{SSPW} \\ \text{RCPW} \end{Bmatrix} = \left(\frac{1-w}{2}\right)^2 \cdot \begin{Bmatrix} \text{SSP1} \\ \text{RCP2.6} \end{Bmatrix} + \frac{1-w^2}{2} \cdot \begin{Bmatrix} \text{SSPM} \\ \text{RCP4.5} \end{Bmatrix} + \left(\frac{1+w}{2}\right)^2 \cdot \begin{Bmatrix} \text{SSP5} \\ \text{RCP8.5} \end{Bmatrix} \quad (1)$$

A statistical optimisation algorithm (Risk Optimiser) was then employed in PAGE-ICE to find the values of $w$ in Equation 1 that result in a 50% probability for the GMST in 2100 to reach the levels consistent with: first, NDCs from the Paris Agreement extrapolated until 2100 (3.3 °C, $w = -0.14$);[71] second, partially implemented NDCs representing an estimated long-term effect of the US's withdrawal from the NDCs (3.6 °C, $w = 0.1$);[72] and third, business as usual projections without the Paris Agreement (4.2 °C, $w = 0.52$).[71] We also added a 2.5 °C target scenario ($w = -0.7$) which is more ambitious than the NDCs but falls short of the 2 °C target.

The 1.5 °C and 2 °C scenarios, defined as having a 50% chance of keeping the GMST rise in 2100 below the 1.5 °C and 2 °C targets based on PAGE-ICE simulations, require extra abatement relative to RCP2.6. They fall outside the range covered by the SSPW and RCPW pairs described above. We, therefore, introduced an additional abatement rate relative to RCP2.6, the same for all the major GHGs represented in PAGE-ICE, and employed Risk Optimiser to find that it is equal to 0.24% per year for the 2 °C target and 4.05% per year for the 2 °C target scenario. Both of these scenarios overshoot their respective targets during the second half of the 21st century and imply negative $CO_2$ emissions thereafter.

All the RCP scenarios in PAGE-ICE are emissions-driven[73], unlike the concentration-driven RCP scenarios that were used in most CMIP5 experiments[17]. We simulated each SSP-RCP pair out to 2300 assuming constant levels of annual emissions, constant GDP growth rates and zero population growth rates beyond 2100. Under each scenario, we ran 100,000 Monte-Carlo simulations in PAGE-ICE to perform sensitivity experiments for the climatic and economic effects of the PCF and SAF.

**Emulator for the nonlinear PCF.** The new dynamic emulator for $CO_2$ and methane emissions from thawing land permafrost is based on simulations from the SiBCASA and JULES LSMs[7,28], forced by multiple CMIP5 and CMIP3 general

circulation models (GCMs) run under a range of climate scenarios out to 2300. The simulated $CO_2$ and methane fluxes from thawing permafrost as a function of time represent the strength and timing of the PCF.

SiBCASA has fully integrated water, energy, and carbon cycles, and a modified snow model to better simulate permafrost dynamics[74]. The soil model separately tracks liquid water, ice, and frozen organic matter at each time step as prognostic variables, accounting for the effects of latent heat[7,75]. SiBCASA separately tracks $CO_2$ and methane emissions. The model was used to make one of the first estimates of future permafrost degradation and global carbon emissions from thawing permafrost[7]. Here we ran multiple projections from 1901 to 2300 starting from the same initial conditions. We spun up the model until the release from permafrost carbon was negligible, ending up with 560 GtC of frozen permafrost carbon in the top three metres of soil[75,76] by initializing the model with the observed values from the Northern Circumpolar Soil Carbon Dataset version 2 (NCSCDv2)[77]. We used the Climatic Research Unit National Centre for Environmental Predictions (CRUNCEP)[78] reanalysis scaled by global climate projections from CMIP5[17]. We chose CMIP5 models that ran both RCP4.5 and RCP8.5 scenarios out to 2300 and that represent a broad range of warming above pre-industrial temperatures: CNRM-CM5, GISS-E2-H, HadGEM2-ES, IPSL-CM5A-LR and MPI-ESM-LR.

The version of JULES used here has an improved representation of physical and biogeochemical processes in the cold regions[79,80]. Competition of vegetation was enabled, allowing the models to determine both their initial vegetation distributions and litterfall, and the response of the vegetation distribution and litterfall to climate change. The profile of soil carbon was spun up until it was in equilibrium with the 1860's climate, giving 738 GtC in the top 3m of soil. Any soil carbon in the permafrost in 1860 was labelled as permafrost carbon and traced throughout the simulation. We assumed that any part of this permafrost carbon which is emitted to the atmosphere is emitted in the form of $CO_2$ only. JULES was forced by climate patterns from the full set of 22 CMIP3 climate model simulations under the RCP2.6, RCP4.5 and RCP8.5 scenarios, extended out to 2300 using the IMOGEN climate emulator[28].

The dynamic emulator of the permafrost carbon emissions is based on a nonlinear first order ODE:

$$\frac{dC}{dt} = \frac{C_{max}}{\tau\,\varphi_\tau(T)} \cdot \left(\frac{\max\left(C_{eq}(T) - C, 0\right)}{C_{max}}\right)^{(1+p)\,\varphi_p(T)} \quad (2)$$

Here $T = AF_p \cdot GMST$ is mean annual permafrost temperature anomaly in year $t$, averaged spatially across the estimated pre-industrial permafrost regions (□C relative to pre-industrial levels); $AF_p$ is the permafrost amplification factor which links $T$ with the GMST anomaly; $C$ is cumulative permafrost carbon emitted since the pre-industrial period as of time $t$ (GtC, either $CO_2$ or methane component); $C_{eq}(T)$ is equilibrium cumulative carbon emitted for a constant permafrost temperature anomaly $T$, expressed as

$$C_{eq}(T) = \min\left(\omega\,\varphi_\omega(T) \cdot T, C_{max}\right); \quad (3)$$

$C_{max}$ is a limit on the maximum possible cumulative emissions determined by the initial carbon stock estimates in SiBCASA (560 GtC) and JULES (738 GtC); $\omega$ (GtC $K^{-1}$) is equilibrium sensitivity of the carbon emissions to permafrost warming; $\tau$ (yr) is the time lag at $t = 0$ (pre-industrial) corresponding to the given $C_{max}$; $p$ is a fixed power that defines the dynamics of how the equilibrium is approached; $\varphi_\omega$ (Equation 3), $\varphi_\tau$ and $\varphi_p$ (Equation 2) are temperature-driven corrections to the parameters $\omega, \tau, p$. All the parameters are assumed to be constant unless they are marked as functions. Equation 2 implies no regeneration of permafrost carbon stocks on the timescales considered[81].

The emulator is calibrated, separately, to the $CO_2$ components of the permafrost emissions simulated by SiBCASA and JULES, and the methane component simulated by SiBCASA. Each combination of a GCM ($m$) and climate scenario ($s$), either in SiBCASA or JULES simulations, produces its own set of optimal equilibrium carbon, lag and power parameters $(\omega, \tau, p)_{m,s}$ that achieves the best emulator fit. The resulting statistics for the $\omega, \tau, p$ parameters is based on the assumptions of equal weights between the GCMs and the scenarios. The corrections $\varphi_\omega, \varphi_\tau, \varphi_p$ (all non-negative) ensure quasi-independence of the $(\omega, \tau, p)_{m,s}$ set as a whole from the scenarios or climate models used. The latter allows us to use these sets of values to construct the corresponding probability distributions for $\omega, \tau, p$ in PAGE-ICE, which are expected to work throughout the simulated range of temperatures. The full technical details of the calibration algorithm and the resulting numerical values for the SIBCASA and JULES emulators are provided in Supplementary Note 2, Supplementary Figs 5–17 and Supplementary Tables 2–6.

The type of a model described by Equation 2 and Equation 3 is often referred to as pursuit curve, and its simpler quasi-linear version ($p = 0$) has been employed for sea level rise emulators previously[82,83]. Even in its simpler form, such a model has never been applied to projected permafrost emissions from process-based simulations of LSMs. The pursuit curve model ensures that there is an equilibrium level of cumulative carbon emissions from permafrost for any given level of warming globally (providing $p > -1$). The dynamic model formulation employed here contains the following layers of nonlinearity: nonlinear response of the equilibrium cumulative carbon to GMST changes, represented by the $\omega\varphi_\omega(T) \cdot T$ term; evolution of the characteristic time lag for cumulative permafrost emissions with the difference between the equilibrium and realised cumulative carbon, represented by $p$ (in the corresponding linear model $p = 0$ and the lag is simply

equal to $\tau$); temperature-dependence in the lag and power parameters, represented by $\varphi_\tau, \varphi_p$; and, saturation of the cumulative carbon emissions due to the permafrost carbon stock exhaustion, represented by $C_{max}$.

The cumulative carbon emissions from the emulators, calibrated separately to SiBCASA and JULES simulations, were averaged with equal weights, both for $CO_2$ and methane, and scaled according to the uncertainty in the observed permafrost carbon stocks[31]. As JULES does not model permafrost methane emissions explicitly, the latter were inferred from its $CO_2$ emissions using observational constraints[84]. The resulting cumulative $CO_2$ and methane emissions from permafrost simulated by PAGE-ICE are plotted in Fig. 7 under the range of scenarios considered.

**Emulator for the nonlinear SAF**. Our nonlinear SAF estimates are based on the ALL/CLR method with atmospheric reflectivity parameterisation[32,33], which uses CMIP5 GCM simulations for atmospheric shortwave radiation fluxes from pre-industrial conditions until either 2100 or 2300 under RCP8.5 scenario (Supplementary Note 3). None of the GCM variables were bias-corrected in order to preserve internal consistency of the sea ice and land snow physics in each model. The statistics of the nonlinear SAF assumes model democracy in the CMIP5 sample used (equal weights for all GCMs).

Applying the ALL/CLR method to the transient GCM simulations produced time series for the global RF associated with the surface albedo changes. These were differentiated with respect to GMST trends over 30-year climatological windows, separately for each model, using linear polynomial fitting to obtain climatologically-averaged SAF in each year. A Savitzky–Golay filter (base period = 31 years; polynomial order = 1) was applied to obtain smooth time series for GMST and SAF. The SAF (both global total and separately for the three main components) was then represented as a function of the GMST rise individually for each model, at which point the multi-model statistics was calculated.

We based the emulator of the global nonlinear SAF on a two-segment approximation described by the following expressions for the SAF, $f(T)$, and the associated RF, $F(T)$:

$$f(T) = \begin{cases} a_0 + a_1 T + a_2 T^2 + \sigma\varepsilon, & T < T_* \\ b_0 + \rho\varepsilon, & T \geq T_* \end{cases}$$

$$F(T) = \int_0^T f(T')dT' = \begin{cases} (a_0 + \sigma\varepsilon)T + \frac{1}{2}a_1 T^2 + \frac{1}{3}a_2 T^3, & T < T_* \\ (a_0 + \sigma\varepsilon)T_* + \frac{1}{2}a_1 T_*^2 + \frac{1}{3}a_2 T_*^3 + (b_0 + \rho\varepsilon) \cdot (T - T_*), & T \geq T_* \end{cases}$$

$$(4)$$

Here $T$ is the GMST anomaly (not to be confused with the permafrost temperature), $T_* = 10\,°C$ is an empirically determined switch between the quadratic and constant SAF segments (Fig. 8), $a_j$ are the coefficients of quadratic polynomial fitting to the multi-model mean global SAF over the $T < T_*$ segment, $b_0$ is average of the multi-model mean global SAF over the $T \geq T_*$ segment, $\sigma(\rho)$ is average of the multi-model mean SD of the global SAF over the $T < T_*$ ($T \geq T_*$) segment, and $\varepsilon = \mathcal{N}(0,1)$. The full technical description of the implementation of the SAF emulator in PAGE-ICE is provided in Supplementary Note 3.

**PAGE-ICE IAM**. PAGE-ICE (v6.22) is based on the PAGE09 IAM[19,20]. It includes several updates both to climate science and economics from IPCC AR5 and literature that followed, as well as several novel developments presented in this paper. The updates are summarised below, with the full technical description provided in Supplementary Note 1, Supplementary Figs 18–23 and Supplementary Tables 7–17.

PAGE and similar IAMs do not model natural climate variability, and therefore each Monte-Carlo run is deterministic in time. This allows us to work with Monte-Carlo generated probability distributions of multiple climatic and economic parameters in any fixed analysis year like 2100, as opposed to taking averages over the 30-year climatological windows (a standard requirement for any climate model data with multiple natural variability cycles). The ranges for all the uncertain parameters in PAGE-ICE are listed in the Supplementary Table 17.

Generic updates in PAGE-ICE: first, adjusted analysis years starting with 2015 (base year), 2020, 2030, 2040, 2050, 2075, 2100, 2150, 2200, 2250 and 2300, allowing for a better representation of the essential long-term processes: permafrost emissions, winter sea ice and land snow decline and melting of the ice sheets; second, updated base year (2015) data for the emissions, temperature, population, GDP-PPP, cumulative permafrost emissions and surface albedo feedback, with uncertainty ranges for most parameters; third, updated set of emissions (RCP) and socio-economic (SSP) scenarios paired according to the RCP-SSP compatibility conditions[22], and modified to cover the range of scenarios in line with the Paris Agreement, as well as the possibility of a reversal of climate policies in the US and globally.

Climate science updates in PAGE-ICE: first, internal dynamic representation of the nonlinear PCF and SAF using emulators based on simulations from multiple CMIP5 and CMIP3 GCMs and SiBCASA and JULES LSMs run under the extended RCP8.5, 4.5 and 2.6 (only JULES) scenarios out to 2300 (see the relevant Methods sections above); second, adjusted transient climate response (TCR), feedback response time (FRT) and ECS parameter ranges according to IPCC AR5 based on CMIP5 models, paleo-records and climate models of intermediate complexity; third, revised $CO_2$ cycle in line with the latest multi-model assessment of the

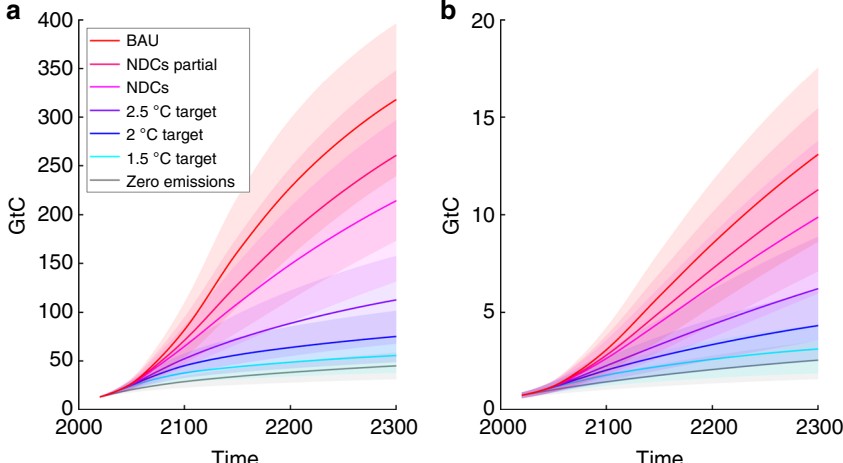

**Fig. 7** Cumulative carbon emissions from the permafrost simulated using the dynamic emulator. Cumulative carbon emissions from thawing land permafrost for **a** $CO_2$ and **b** methane components simulated by the new statistical emulator of SiBCASA and JULES (equal weighting) under the chosen range of climate scenarios until the year 2300 (solid lines: mean; shaded areas: ±1SD). 100,000 Monte-Carlo runs of PAGE-ICE. Units: GtC. Note the difference in the Y-axis scale between the plots. Source data are provided as a Source Data file

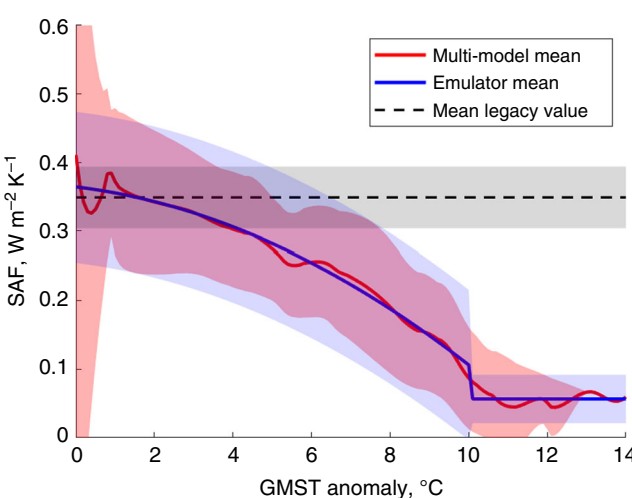

**Fig. 8** Emulator of the global surface albedo feedback and its legacy value. Global SAF as a function of the GMST rise relative to pre-industrial conditions obtained from 16 CMIP5 GCMs using the ALL/CLR method. Red line: multi-model mean; shaded red area: ±1SD; blue line and shaded area: mean and ±1SD of the two-segment emulator. The dashed line and grey shaded area show statistical mean and ±1SD of the SAF averaged between pre-industrial conditions and the level of warming corresponding to the 2xCO2 ECS experiment (mean value of 2.8 °C, 5–95% range of 1.7 °C–4.2 °C according to IPCC AR5). Source data are provided as a Source Data file

atmospheric $CO_2$ response function;[85] fourth, improved GMST equation using a better numerical scheme for finite analysis periods; fifth, CMIP5-based amplification factors for the regional temperatures; sixth, changes in the implementation of the regional sulphate cooling: sulphates now add to the global forcing and affect the regional temperatures implicitly through the CMIP5-based amplification factors (their RF is not included in the regional temperature equation directly due to the complexity of climatic response to regional RFs, which requires regional climate sensitivities to be introduced;[86] seventh, approximately halved indirect sulphate cooling effect; eighth, fat-tailed distribution for the sea level rise (SLR) time lag (at the lower values end) to account for the possible acceleration in the discharge from the West Antarctica and Greenland ice sheets[87–90].

Economics updates in PAGE-ICE: first, new economic impact function based on the recent macro-econometric analysis of the effect of historic temperature shocks on economic growth in multiple countries by Burke et al.[46], projected onto the 8 major regions of the PAGE model using population-weighted temperatures,

and adapted to fit with the single year consumption-only approach for climate impacts used in PAGE; second, considerably downscaled saturation limit for the impacts; third, modified uncertainty range for the BaU scenario, which is used as a reference point for calculating the abatement costs, covering the range roughly between RCP6.0 and a pathway exceeding RCP8.5;[23] fourth, revised present-day marginal abatement cost (MAC) curves[64], technological learning rate ($CO_2$ only)[65] and autonomous technological change based on energy efficiency improvements;[91] fifth, significantly downscaled discontinuity sector, which now accounts only for socio-economic tipping points such as pandemics, mass migration and wars, as well as possible other tipping points in the climate than permafrost, sea ice, land snow and lea level rise from ice sheets (the catastrophic loss of the ice sheets has been moved to the fat-tailed distribution in the sea level rise module); sixth, reduced tolerable temperature rise that gives no chance of a discontinuity; seventh, significantly decreased time constant of a discontinuity in line with its new interpretation; eighth, focus on autonomous adaptation as part of the Burke et al. economic impact function, with planned adaptation restricted to SLR impacts.

**Climate model data**. The complete lists of the CMIP5 and CMIP3 models used in the study are provided in Supplementary Tables 18 and 19.

**Image processing**. The Figures were plotted using Matlab R2018a, IDL (Fig. 5) and Palisade Risk 7.5 (Fig. 6). We used Matlab's Savitzky–Golay smoothing for the SAF results from CMIP5 (Fig. 2) and Piecewise Cubic Hermite Interpolating Polynomial (PCHIP) interpolation for the time-series results from PAGE-ICE.

## Data availability
All data generated or analysed during this study are included in this published article and its Supplementary Dataset files, with exception of the publically available CMIP datasets acknowledged below.

## Code availability
The PAGE-ICE model (v6.22) and the associated pre- and post-processing computer codes are included in the Supplementary Code files. The SiBCASA and JULES models are managed, respectively, by expert teams at the National Snow and Ice Data Centre (US) and at the UK Met Office, and are not included in this publication due to their complexity.

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

## Acknowledgements

This work is part of the ICE-ARC project funded by the European Union's 7th Framework Programme, (grant 603887, contribution 006). D.Y. received additional funding from ERIM, Erasmus University Rotterdam, and Paul Ekins at the ISR, University College London. K.S. was funded by NSF (grant 1503559) and NASA (grants NNX14A154G, NNX17AC59A). E.J. was funded by the NGEE Arctic project supported by the BER Office of Science at the U.S. DOE. Y.E. was funded by the NSF (grant 1900795). E.B. was supported by the UK Met Office Hadley Centre Climate Programme funded by BEIS and DEFRA. Publication of this article was funded by Lancaster Environment Centre, the University of South Florida St. Petersburg's Open Access Publication Fund, NSF (grant 1900795) and NASA (grants NNX14A154G, NNX17AC59A). We thank the five anonymous referees for providing wide-ranging critical comments that helped improve the paper considerably. We are also grateful to multiple colleagues from the ICE-ARC consortium and beyond for a number of useful discussions that contributed to shaping this study, including Jeremy Wilkinson, Peter Wadhams, Michael Karcher, Frank Kauker, Rüdiger Gerdes and Andy Jarvis. Special thanks go to Michael Winton for providing the original ALL/CLR script. We also acknowledge the World Climate Research Programme's Working Group on Coupled Modelling, which is responsible for CMIP, and we thank the climate modelling groups (listed in Supplementary Tables 18 and 19) for producing and making available their model output. For CMIP the U.S. DOE's Program for Climate Model Diagnosis and Inter-comparison provides coordinating support and led development of software infrastructure in partnership with the Global Organization for Earth System Science Portals.

## Author contributions

D.Y., G.W. and C.H. conceived the research; D.Y. and C.H. created the PAGE-ICE model and ran the simulations; D.Y. developed and calibrated the permafrost and albedo feedback emulators; K.S. and E.J. designed and ran SiBCASA simulations; E.B. designed and ran JULES simulations; K.R.C., F.I.S. and D.Y. adapted the ALL/CLR script for the albedo feedback; F.I.S., K.R.C., K.S., P.Y. and Y.E. processed climate model data; All authors provided input on the scientific and policy matters and contributed to the writing of the paper.

## Additional information

**Competing interests:** The authors declare no competing interests.

