## [Peer Review File · Nature Communications]

Reviewers' comments:

Reviewer #1 (Remarks to the Author):

The authors present an updated version of an integrated assessment model (IAM) that includes two new non-linear Arctic feedbacks: the permafrost carbon feedback (PCF) and the sea-ice albedo feedback (SIAF), the strength of which depends on the amount of sea-ice present. While the latter is naturally taken into account in climate models the IAM emulates (including its varying strength), the PCF was not represented in the ESMs participating to CMIP5, which is the model generation IAMs currently emulate. Taking into account the state-dependency of the SIAF in an IAM is a novelty as far as I can judge as a climate scientist not particularly familiar with IAMs. The authors then assess interactions between these feedbacks in a broad range of greenhouse gas emission scenarios until 2300 and evaluate the economic impacts of these interactions as well as, more generally, the economic cost of the different scenarios, including the cost of the US's withdrawal from the Paris agreement.

Here the paper loses somewhat its focus. What are the main results the authors want to convey? It is somewhat unclear to the reader whether the main results concern the nonlinear interactions between the Arctic feedbacks and their economic implications or, more generally, the updated economic consequences of the different scenarios.

I personally feel somewhat unwell with the way the economic costs of the different scenarios are presented; does it really make sense to present integrated economic costs over 300 years? Of course, integrating global numbers over 300 years yields big numbers, but is there really a point in doing so, given that technical, political and societal changes over the next 300 year might be, at least, quite tricky to predict? It would sound more sober and more pertinent to me to present annual costs on a human time horizon (say, until 2100 or so). However, as a climate scientist, I will focus more on the more strictly climate-related aspects of the paper.

The economic impact of the PCF until 2300 has been taken into account by essentially the same group of authors in a paper published recently (and, to prevent any misunderstandings, that paper is of course correctly referred to in the present paper). The varying strength of the PCF, including a possible depletion of the reservoir and the fact that the relative impact of the PCF is stronger for lower emission scenarios, has also been known before. The novelty of this paper, as also acknowledged by the authors, lies more in the fact that the varying strength of the SIAF is taken into account (the feedback vanishes once there is no more sea-ice). The authors do this by modulating the prescribed equilibrium climate sensitivity (ECS) and feedback response time of the upper ocean (FRT), which are global key parameters of the IAM.

The results show that the interactions between these feedbacks are, on average, fairly weak. The essential reason is that the SIAF itself is much weaker than the PCF. In most scenarios, the economic impact of the coupling between these two Arctic feedbacks is far below half a percent of the economic cost of the climate change. This is a very marginal impact, and as such, even if the

integration over 300 years yields substantial total cost, I doubt that the result is of particular interest in term of economical impact.

I am not convinced either that the scientific progress as such is substantial enough to warrant publication in this journal. The work focuses on the PCF and the SIAF. The SIAF is taken into account, as stated before, by modulating the ECS and the FRT. The authors could have taken a much broader approach by using the CMIP5 archive and additional long-term climate simulations to parameterize the state-dependency of the climate sensitivity simulated by these models. In terms of high-latitude feedbacks, this would then, by construction, also have taken into account the (continental) snow albedo feedback (SAF) – I see no particular reason not to take into account the SAF. (Changing continental ice sheet extent would still have been neglected if CMIP5 had been used.) Quite certainly global cloud feedbacks and their uncertainties would still have dominated the global picture, yielding very weak variations in ECS. In my sense, focusing on high latitude feedbacks is a bit too partial an approach, yielding a skewed picture of the reality. This seriously compromises the interest of this work.

Some specific points:

- Abstract: “.. the permafrost feedback is significant...”: It might be better to use “substantial” instead of “significant” which has a very specific meaning
- The author use the word “trillion”, which is of course correct but still misleading because the same word has a different meaning in some other languages. It might be useful to indicate at least once how much this is in the authors’ minds (that is, use the scientific notation 10^{12} (if I’m not mistaken) to prevent misunderstandings)
- The abstract does not indicate how strong these nonlinear interactions are (they are weak...) – this is at odds with the title which insists on the non-linearity as the key scientific progress presented.
- “Private correspondence” is rather often referred to in this paper. Try to reduce, please, if possible.
- Introduction: The paragraph “The Paris agreement of December 2015...” does not really belong here and adds little to the understanding of the paper.
- “more complicated physical models”: “more complex” might be a better word here.
- Throughout the paper, it is not particularly clear what “permafrost emissions” really are. Are these emissions of CO₂ and CH₄ from areas where permafrost has been present at some reference time (presumably today), or more specifically emissions from soils that were frozen at that time, that is, excluding soils currently in the active layer above permafrost?
- Would it be possible to use results from other land surface models, not only SiBCASA? Several GCMs are used, so why not also GHG emissions from other LSMs which are highly uncertain?
- The step approximation for the SIAF looks a bit like overkill. The essential behavior of the SIAF seems to be a constant value of about 0.15 Wm⁻²K⁻¹ until +7K from today, and then 0. Wouldn’t this be much more parsimonious, given the large uncertainties?

- “The GMST changes from PCF and SIAF are smaller than the underlying uncertainty in the baseline climate projections”. This, again, shows that the results do not necessarily warrant a high-profile publication, IMHO, although they are statistically significant.
- “... leading to a growth of the permafrost area and refreezing of some of the thawed carbon”. The description of the PCF is not particularly clear in terms of a necessary distinction between thawed and emitted carbon. Does the sentence “This reduces the amount of thawed carbon, causing a decrease in the PCF strength” mean that thawed permafrost carbon is explicitly represented in the model, or is it conceptually the same as “emitted carbon”? Looking at the equations in the methods section, it seems that the process of carbon accumulation and emission are represented by the same equation, although the processes (and the timescales) are very different: carbon accumulation in permafrost areas (and anywhere else) is a very slow process, while carbon emission can be relatively fast (even if it still takes decades to centuries). Some strange features of the strong mitigation scenarios as calculated by the IAM presented here might be explained by this unrealistic symmetry in the model formulation and response.
- “For the High fossil fuels use scenario...”: for clarity, you might add here that, in addition, the PCF is exhausted in this scenario after 2250.
- Figure 6: One wonders what the NPV for very low GMST changes is in the “default impact” scenario. Should probably be 0 but the figure (which invites to extrapolate by eye) does not suggest it.
- “These results do not include other known climatic feedbacks...”: The whole sentence is not understandable, at least not to me, probably as a result of a copy-paste that went wrong. It looks like two fragments of independent lines of thought.
- The two sentences following immediately after (“Science is still limited... additional losses from the two Arctic feedbacks.”) are also unclear. Why should an underestimate be likely? Why not an overestimate?
- “For most scenarios... PCF... impacts are greater for higher emissions scenarios.” Of course they are, but relative impacts are more interesting.
- The ECS in CMIP exercises is typically calculated using the abrupt4xCO2 experiment (e.g. Andrews et al., GRL 2012, doi:10.1029/2012GL051607). At 4xCO2, there is not much sea-ice left. This means that the ECS used by default should take this sea-ice depletion into account. This does not appear clearly enough in the paper (but it does not annihilate the progress made by taking into the variability of the SIAF, higher in the beginning than at the end of high emission scenarios).
- The summary paragraph starting with “The climate policy implications of the Arctic feedbacks are profound.” comprises in a nutshell the main troubles I have with the paper. The points mentioned in the paragraph show are either that the results presented here are not new or show that these results do not induce a substantially renewed view of the problem. It is not new that the PCF, not taken into account in CMIP5 climate models (and thus in most IAMs) requires additional mitigation efforts. It is no new that a better understanding of the complex processes in the Earth’s climate system is required. The paper shows (again) that the PCF and the SIAF become less important at higher GMST changes, which is somewhat at odds with the statement in this paragraph

that “triggering these planetary feedbacks might accelerate the pace of climate change”. And it is not clear to me why the results presented here should show why current IAMs “likely... underestimate the total economic effect of climate change”: the effect of the PCF (increased cost) has been shown by the authors before and the results here show reduced cost in the long term. They do not show any clear results concerning other tipping points that warrant the assessment that the economic costs of climate change will be higher than current assessments indicate.

- “Further interdisciplinary research is needed to improve our understanding of the complex nonlinear interactions both in the Earth’s climate and socio-economic systems.” IMHO, there is first a particular need for disciplinary research to improve the understanding of nonlinear interactions within the Earth’s climate and within the socio-economic systems separately.

- Some remarks on the methods section:

o Please define “warming degree-years”

o “The nonlinear Equation 1 and Equation 2 are solved analytically...” Really? Not numerically?

o The parameterization of the SIAF does not necessarily appear robust with respect to the choice of the ESMs. It might be more reasonable to use a more simplified parameterization based on only two or three eras.

o The scientific literature clearly distinguishes between the Charney-type ECS and the Earth System ECS, which basically includes feedback strength variations as those discussed here. The authors might want to reflect this discussion here.

Reviewer #2 (Remarks to the Author):

This paper provides new and interesting insights on the climatic and economic implications of two important Arctic related process, i.e. the permafrost carbon feedback (PCF) and the sea-ice albedo feedback (SIAF). Recently, different studies have captured and explored the implications of these two elements in isolation and/or in exogenous way in IAM models. However, most of these studies have not disentangled, for example, the level of SIAF that is typically included in the equilibrium climate sensitivity (ECS) parameter. The novelty of this paper is that both elements are, for the first time, considered in combination and, also that both effects are endogenously and transparently captured within the model. This is done via coupling an emulator of both physical process within a well-known IAM model (PAGE model), which was used for the Stern Report.

The calculations seem to have been done competently and the result is written up in a transparent way. The implications of this paper will be of the interest of the scientific community on climate

change. Moreover, and given the rapid changes that we are seeing in the Arctic, I think that studies like this will be relevant for climate policy and for policy makers as it helps to improve our understanding on the future impacts from climate change and also of the risk associated to crossing some critical thresholds. Therefore, I think this work is acceptable for publication if the following questions are clarified:

- 1) Scenario zero emissions. Add another scenario where GHG emissions are immediately reduce to zero forever. Although this scenario is not realistic it would give to this paper an interesting new insight which is to know the increase in the temperature we can expect due to the inertia of the climate system with and without the PCF and SIAF. This scenario will also be very useful in order to compare the result obtained in this paper with the PAGE-ICE model with those from the IPCC AR5 report (Figure 1, FAQ 12.3, Chapter 12, WG1).
- 2) Scenario NDCs no-US. Although I like the structure of the scenario, the “NDCs no-US” scenario is bit out of scope when doing projections to 2100 and 2300. The scenario can be maintained but I would recommend not to highlight it in the abstract and better to concentrate more the attention on the 1.5 and 2-degree scenarios which are the policy relevant. Finally, notice that after the COP 21 and the Paris Agreement the contributions are not anymore “intended”.
- 3) Models selected for emulator calibration. It is not clear (not for PCF nor for the SIAF) which is the criteria used for selecting some of CMIP5 models versus others. This is crucial for the calibration of the emulators and, therefore, for the results of the paper. Please, explain briefly in the manuscript even if a more technical detail is needed or included in the SM.
- 4) Uncertainty analysis. The paper results are based on a Monte-Carlo generated probability distribution of “multiple climatic and economic parameters”. Please, explain which are those parameters and what are the distribution selected for them.
- 5) SIAF process. The figure 2 of the paper is not explained. Also, I am not convinced about the step-approximation followed (figure 8), why eight distinct eras? which is the foundation for this, is statistically-related or ice-process-related or both? It is not clear. Also, it is quite strange that the MPI-ESM-LR model match/fit 100% to this, how this could be possible?
- 6) Cumulative RF: Please, explain the concept of “cumulative” radiative forcing of Figure 5 and how do you calculate this.
- 7) SIAF and winter sea ice. You say that “constant SIAF greatly overestimates the GMST increases compared to the non-linear SIAF beyond 2150, which is due to the loss of the winter sea ice”. The role of winter sea ice is not totally clear in this sentence. If I understand this correctly, the issue is that there is a saturation point (when winter sea ice is melt) where the extra warming cannot increase more (there is no sea ice anymore) and that constant SIAF does not capture this maximum level, right?
- 8) Data of Table 1. Please check that the all data of this table is correct and also that data reference in the text match with the table. For example, the total economic impact of the 1.5 scenario (\$566tr) is higher than the 2°C scenario (\$476tr), is this correct? Are the extra-damages in

PAGE model greater than the extra-mitigation cost of going from 2°C to 1.5°C? I guess that Burke's estimates would change this.

9) Coupling effect: Also explain better in the text when you refer in Table 1 to the "coupling effect", it is not clear. The coupling effect is to me very relevant and the results of the table could also be used to connect with the conclusion and further research section when you mention that your results are likely an underestimation as some processes are not captured. Although the SIAF effect seems small compared to the PCF, I wonder if due to its coupling effect with others major systems (Greenland, THC, etc.) this effect could be still critical.

Reviewer #3 (Remarks to the Author):

This paper looks at the economic impact of including both the permafrost carbon feedback and the sea ice albedo feedback in an integrated assessment model (IAM). The work builds on the paper by Hope and Schaefer (2016) which looks at the financial consequences of including just permafrost CO₂ and CH₄ release in an IAM. It includes more scenarios (particularly the 1.5 and 2.0 degree stabilisation scenarios) but a relatively limited set of uncertainties. The sea ice albedo feedback has previously been included in an IAM but in a simpler form. There are some relatively small interactions between the permafrost feedback and the sea ice feedback.

In the abstract they pick out one key result regarding the cost of the two Arctic feedbacks in 2300 of the USA withdrawal from the Paris agreement. Studies of the permafrost carbon feedback typically go out beyond 2100 because at the longer time scales the carbon has had more time to decompose and be released and have a notable impact. However, in the abstract I might also pick out another key result for the lower temperature targets.

Results from SiB-CASA are used to calibrate the permafrost carbon model. The setup of SiB-Casa was originally described by a Schaefer et al. (2011) paper - further details are still needed as to how the methane release from the permafrost is estimated.

The estimate of permafrost carbon lost as CO₂ and CH₄ is high compared with previous work (for example Burke et al. 2017; Schneider von Deimling et al., 2015) Whilst this is not necessarily wrong, it probably needs to be discussed. If all of the permafrost carbon is thawed by 2300 the majority of it will have to have decomposed to emit 500 Gt C by 2300. This decomposition process is relatively slow if it is just related to active layer deepening. Therefore, I think these high losses need to be

justified more carefully. In addition, the temperature responses to the permafrost carbon feedback in Figure 4a are different to those found in Burke et al., 2017, Figure 7. Is this caused by the additional inclusion of methane?

I feel the uncertainties in the Arctic feedbacks need to be significantly expanded on. There are many uncertainties in the release of permafrost carbon which are not even discussed. Also the uncertainties need to feed through into the economic impact.

The most relevant timescales of these economic impacts need to be explained more carefully. There is both an instantaneous impact and a longer term impact. This could be made clearer and is definitely of interest.

Economics - A "discounted economic impact" is a little confusing term.

Schneider von Deimling T, Grosse G, Strauss J, Schirrmeister L, Morgenstern A, Schaphoff S, Meinshausen M, Boike J. Observation-based modelling of permafrost carbon fluxes with accounting for deep carbon deposits and thermokarst activity. *Biogeosciences*. 2015;12(11):3469-88.

Hope, C. and Schaefer, K., 2016. Economic impacts of carbon dioxide and methane released from thawing permafrost. *Nature Climate Change*, 6(1), p.56.

Burke, E. J., Ekici, A., Huang, Y., Chadburn, S. E., Huntingford, C., Ciais, P., Friedlingstein, P., Peng, S., and Krinner, G.: Quantifying uncertainties of permafrost carbon-climate feedbacks, *Biogeosciences*, 14, 3051-3066, <https://doi.org/10.5194/bg-14-3051-2017>, 2017.

SCHAEFER, K., ZHANG, T., BRUHWILER, L. and BARRETT, A. P. (2011), Amount and timing of permafrost carbon release in response to climate warming. *Tellus B*, 63: 165–180. doi:10.1111/j.1600-0889.2011.00527.x

Response to Referees

We are grateful to the Reviewers for their constructive comments, and welcome the opportunity to revise and resubmit the manuscript.

To address the significant points raised by the Reviewers, we undertook substantial revisions of several key components of the manuscript, focusing on improving the emulators of the nonlinear Arctic feedbacks, refining the estimates of the economic impact of climate change in the PAGE-ICE model, and making the key contributions and implications of this work clear in the text.

Here we respond to the specific points raised by the Reviewers, listed in black italic.

Reviewer #1 (Remarks to the Author)

The authors present an updated version of an integrated assessment model (IAM) that includes two new non-linear Arctic feedbacks: the permafrost carbon feedback (PCF) and the sea-ice albedo feedback (SIAF), the strength of which depends on the amount of sea-ice present. While the latter is naturally taken into account in climate models the IAM emulates (including its varying strength), the PCF was not represented in the ESMS participating to CMIP5, which is the model generation IAMs currently emulate. Taking into account the state-dependency of the SIAF in an IAM is a novelty as far as I can judge as a climate scientist not particularly familiar with IAMs. The authors then assess interactions between these feedbacks in a broad range of greenhouse gas emission scenarios until 2300 and evaluate the economic impacts of these interactions as well as, more generally, the economic cost of the different scenarios, including the cost of the US's withdrawal from the Paris agreement.

Here the paper loses somewhat its focus. What are the main results the authors want to convey? It is somewhat unclear to the reader whether the main results concern the nonlinear interactions between the Arctic feedbacks and their economic implications or, more generally, the updated economic consequences of the different scenarios.

We agree with the Reviewer and revised the main text to state our objectives more clearly. We also modified the summary and abstract to clearly state our primary conclusions.

The main goal of our study is to investigate nonlinear transitions in the PCF and SIAF, and their respective interactions with and impact on global climate and economy using the new version of the PAGE IAM. While several elements of these nonlinear processes and the respective climatic and economic impacts have been studied before (Winton, 2006, 2008; Hope and Schaefer, 2016; Burke et al., 2017, 2018; Gonzalez-Eguino et al., 2016, 2017), we believe that our work provides the most comprehensive analysis to date for both Arctic feedbacks. As such, it contributes to quantifying the uncertainties surrounding the three corresponding Arctic tipping elements and their global impacts, as proposed by Lenton et al. (2008) and Schellnhuber et al. (2016). We discuss the significance of our findings and of the proposed new methodologies below.

The interaction between the PCF and SIAF themselves is of a lesser importance, as the Reviewer correctly pointed out, and therefore we do not describe it in the revision, focusing on statistical significance of the effects of the PCF and SIAF instead. We modified the main text accordingly.

I personally feel somewhat unwell with the way the economic costs of the different scenarios are presented; does it really make sense to present integrated economic costs over 300 years? Of course, integrating global numbers over 300 years yields big numbers, but is there really a point in doing so, given that technical, political and societal changes over the next 300 year might be, at least, quite tricky to predict? It would sound more sober and more pertinent to me to present annual costs on a human time horizon (say, until 2100 or so). However, as a climate scientist, I will focus more on the more strictly climate-related aspects of the paper.

This is a valid argument that applies both to climate models' and IAMs' projections. The scenarios are extended out to 2300 to capture the effects of multiple slow physical processes including the PCF and the loss of the winter sea ice under high emissions pathways. Earlier work has shown that a very substantial fraction of the total discounted economic effects occur after 2100, so an analysis that ended in 2100 would be seriously incomplete. While very long horizons like this may appear irrelevant from the point of view of the actual socio-economic processes, the well-established technological, demographic and resource constraints (Riahi et al., 2017; Christensen et al., 2018) imply that the range of scenarios is still plausible beyond the 21st century (Yumashev, 2018). We reflected on this in the opening section.

The economic impact of the PCF until 2300 has been taken into account by essentially the same group of authors in a paper published recently (and, to prevent any misunderstandings, that paper is of course correctly referred to in the present paper). The varying strength of the PCF, including a possible depletion of the reservoir and the fact that the relative impact of the PCF is stronger for lower emission scenarios, has also been known before. The novelty of this paper, as also acknowledged by the authors, lies more in the fact that the varying strength of the SIAF is taken into account (the feedback vanishes once there is no more sea-ice). The authors do this by modulating the prescribed equilibrium climate sensitivity (ECS) and feedback response time of the upper ocean (FRT), which are global key parameters of the IAM.

The novelty of our work is not just in taking into account the varying strength of the SIAF and modelling it in an IAM, as suggested by the Reviewer. Our paper introduces new methodologies for emulating nonlinear transitions both in the PCF and SIAF, and uses PAGE-ICE IAM to perform a comprehensive statistical analysis of the resulting climatic and economic impacts, with implications for the Paris Agreement targets. PAGE-ICE is a substantial analytical update to the previous version of the PAGE model used in the earlier paper on the economic effect of the PCF by Hope and Schaefer (2016). It includes internal nonlinear dynamic representation of the PCF and SIAF, along with a number of updates to the science and economics in line with the latest literature (see Methods and Supplementary Materials).

Our work also provides a possible new framework for quantifying the climatic and economic effects of a number of other known tipping elements in the climate system (beyond the SIAF

and PCF), which is essential for putting the far-reaching climate policy commitments like the 1.5°C target on a solid scientific basis.

The results show that the interactions between these feedbacks are, on average, fairly weak. The essential reason is that the SIAF itself is much weaker than the PCF. In most scenarios, the economic impact of the coupling between these two Arctic feedbacks is far below half a percent of the economic cost of the climate change. This is a very marginal impact, and as such, even if the integration over 300 years yields substantial total cost, I doubt that the result is of particular interest in term of economical impact.

First, the interaction between the PCF and SIAF is no longer a subject of the revision, as we already mentioned earlier.

Second, as part of the revision, we carried out extensive new analysis of the SIAF using a wider pool of CMIP5 models, and remodelled the SIAF emulator to make it statistically robust. The resulting economic impact of the SIAF increased substantially as a result, reaching nearly 4% of the total effect of climate change under the NDCs scenario. The highest impact of the PCF is achieved under the same scenario and amounts to around 7% of the total effect of climate change.

Finally, the use of the 300-year period has also been discussed above. Given the Reviewer's background climate science, it is worth clarifying the technicalities of using long-term horizons for economic calculations. We do not just aggregate the economic impacts over the 300-year period, but also discount them in line with a standard methodology adopted from finance, which is common in climate policy assessments based on cost-benefit analysis. The main indicator employed in the analysis is the Net Present Value (NPV) of the economic effect of climate change and, specifically, of the Arctic feedbacks. It uses equity weighting and a pure time preference rate to perform the discounting (Anthoff et al., 2009). The effect is to increase the valuation of impacts in regions that are poorer than the focus region (EU) in the base year, and decrease the valuation of impacts in regions that are richer. In addition, the further in future the impacts are, the less they contribute to the NPV calculation, making the result very different from a simple aggregation. Despite their reduced weight, the impacts that are set to occur in the 22nd and 23rd centuries make a substantial contribution to NPV of the total economic effect. We added these considerations to the Supplementary Materials.

I am not convinced either that the scientific progress as such is substantial enough to warrant publication in this journal. The work focuses on the PCF and the SIAF. The SIAF is taken into account, as stated before, by modulating the ECS and the FRT. The authors could have taken a much broader approach by using the CMIP5 archive and additional long-term climate simulations to parameterize the state-dependency of the climate sensitivity simulated by these models. In terms of high-latitude feedbacks, this would then, by construction, also have taken into account the (continental) snow albedo feedback (SAF) – I see no particular reason not to take into account the SAF. (Changing continental ice sheet extent would still have been neglected if CMIP5 had been used.) Quite certainly global cloud feedbacks and their uncertainties would still have dominated the global picture, yielding very weak variations in

ECS. In my sense, focusing on high latitude feedbacks is a bit too partial an approach, yielding a skewed picture of the reality. This seriously compromises the interest of this work.

We respectfully disagree. We identify a number of key contributions:

- (i) The extensive analytic updates in PAGE-ICE, combined with the integration of nonlinear representation of the PCF and SIAF, provide a substantial improvement on existing IAMs used by economists and climate policy analysts;
- (ii) The results of this improved estimation provide substantial new insights into the economic viability of climate policy estimates. For instance, our results show that the targets of +2.5°C warming and below, relative to pre-industrial conditions, are more economically efficient than higher warming levels. This is significantly lower than existing estimates (Hope, 2015), which found a mean GMST rise of over 3°C to be economically optimal;
- (iii) Furthermore, as we mentioned earlier, our work on emulators of the nonlinear Arctic feedbacks introduces a possible new framework for quantifying the climatic and economic effects of a number of nonlinear feedbacks (beyond the SIAF and PCF) in the climate system, which is valuable in robustly assessing far-reaching climate policy commitments like the 1.5°C target.

Eventually, we plan to include emulators for all the global feedbacks in PAGE IAM to make a “planet emulator”. This is an important future research project, but one that we cannot accomplish in a single paper. Here we want to focus on the high latitude feedbacks. The high latitude feedbacks result in warming at twice the global average rate, indicating climate change is hitting the high latitudes first and hardest, while also causing additional climatic and economic impacts globally. This global perspective, and the possibility to use the new methodology to create a comprehensive planet emulator, makes our results very relevant. We believe this work has clear potential to influence both climate science and policy, which justifies publication in a high impact journal.

To address the specific point regarding the snow albedo feedback (SAF), we performed an additional analysis of CMIP5 simulations out to 2300 and found that unlike the SIAF, the SAF is largely constant throughout the projected warming range under the RCP8.5 scenario extended out to 2300. As the constant SAF is already included in the 2xCO₂ ECS parameter in PAGE-ICE, its explicit addition on par with the PCF and SIAF appears unjustified. Furthermore, the SAF also features in the PCF emulator implicitly through the temperature amplification factor in the permafrost regions. The details are given in Supplementary Materials.

Finally, the large uncertainty in climate warming caused by clouds and aerosol parametrization is an established issue. Of the two most recent studies on the subject that were based on observational constraints (Cox et al., 2018; Brown et al., 2018), one matched closely with the ECS parameterisation from IPCC AR5 adopted in PAGE-ICE, suggesting that our climate projections are robust. We reflected on this in the main text.

Some specific points:

- Abstract: “.. the permafrost feedback is significant...”: It might be better to use “substantial” instead of “significant” which has a very specific meaning

We modified the text accordingly.

- The author use the word “trillion”, which is of course correct but still misleading because the same word has a different meaning in some other languages. It might be useful to indicate at least once how much this is in the authors’ minds (that is, use the scientific notation 10^{12} (if I’m not mistaken) to prevent misunderstandings)

While using the term “trillion” may not be common in climate science, it is widely used in economics. Likewise, “ 10^{12} ” would most likely confuse the economists and policy-makers. Therefore, we deemed it necessary to keep using “trillion” in the text and added a footnote in the relevant section explaining that 1 trillion = 10^{12} .

- The abstract does not indicate how strong these nonlinear interactions are (they are weak...)
– this is at odds with the title which insists on the non-linearity as the key scientific progress presented.

As mentioned in our earlier comments, the revised version of the paper does not focus on the interaction between the PCF and SIAF themselves. The key contribution of the paper is in providing a comprehensive estimate of the impacts of the two nonlinear feedbacks on the global climate and economy under a range of climate scenarios consistent with the Paris Agreement, including the statistical significance of their individual and combined effects. We modified the Abstract to reflect on this.

- “Private correspondence” is rather often referred to in this paper. Try to reduce, please, if possible.

We removed “private correspondence” and added published references where appropriate.

- Introduction: The paragraph “The Paris agreement of December 2015...” does not really belong here and adds little to the understanding of the paper.

We removed this paragraph, while also making it clear elsewhere in the text how our work is relevant for the current climate policy issues.

- “more complicated physical models”: “more complex” might be a better word here.

We amended this.

- Throughout the paper, it is not particularly clear what “permafrost emissions” really are. Are these emissions of CO₂ and CH₄ from areas where permafrost has been present at some reference time (presumably today), or more specifically emissions from soils that were frozen at that time, that is, excluding soils currently in the active layer above permafrost?

Permafrost carbon emissions analysed here are methane and CO₂ fluxes (through plant (autotrophic) and microbial (heterotrophic) respiration) that result from the thaw of soil's old frozen carbon in the permafrost regions. The areas where the emissions occur move northwards according to the simulated level of warming under a specified climate scenario, which is modelled implicitly by the PCF emulator.

- Would it be possible to use results from other land surface models, not only SiBCASA? Several GCMs are used, so why not also GHG emissions from other LSMs which are highly uncertain?

Using the SiBCASA output for the PCF emulator produces reasonable estimates of carbon fluxes that fall roughly in the middle of available estimates from other LSMs, and thus is a justified representative for the range of the available models (Schaefer et al., 2011, 2014; Schuur et al., 2015; McGuire et al., 2018). SiBCASA is also part of the MsTMIP model inter-comparison project (Huntzinger et al., 2013), and has been used to prepare several benchmarks for LSM validations as part of the PBS project (“A Permafrost Benchmark System to evaluate permafrost models”).

The uncertainties captured by the PCF emulator based on SiBCASA represent variations in input climate simulated by the multiple ESMs under different scenarios. This is a valid approach since the dominant source of uncertainty in the PCF is the uncertainty in the projected climate ($\pm 34\%$), while the second largest source of uncertainty is the amount of carbon in permafrost ($\pm 15\%$) (Schaefer et al., 2011).

We reflected on these points in the text.

Finally, including outputs from other LSMs would change the focus of the paper towards investigating the uncertainty of LSM model outputs and its implications for the parameterisation PCF emulator. While this is a worthy topic for a follow-up study, we deem this to be out of the scope of the present work.

- The step approximation for the SIAF looks a bit like overkill. The essential behavior of the SIAF seems to be a constant value of about 0.15 Wm⁻²K⁻¹ until +7K from today, and then 0. Wouldn't this be much more parsimonious, given the large uncertainties?

We agree with first point raised by the Reviewer. To address it, we carried out extensive new analysis of the SIAF using a wider pool of nine CMIP5 models and including their historic simulations in addition to RCP8.5 projections, and remodelled the SIAF emulator to make it statistically robust. The shortlisted models perform well against historic sea ice records from

the satellite datasets, and produce monthly sea ice projections that are not characterised as outliers in the multi-model CMIP5 sample.

The revised estimates for the multi-model SIAF have two distinct peaks associated with the loss of the late summer and early summer sea ice, occurring at around 1°C and 6°C of warming globally relative to pre-industrial conditions (Figure 2). The two peaks are implemented as jumps in the radiative forcing (RF) in the SIAF emulator, with constant SIAF in between. The number of uncertain parameters in the emulator has been reduced from 14 to 5 (4 of them statistically independent, and the other one expressed as their function), while the number of distinct “eras” has been reduced from 8 to 3. This provides a parsimonious but robust interpretation of the uncertainties in the selected pool of climate models.

- *“The GMST changes from PCF and SIAF are smaller than the underlying uncertainty in the baseline climate projections”. This, again, shows that the results do not necessarily warrant a high-profile publication, IMHO, although they are statistically significant.*

All climate feedbacks are small compared to climate change driven by anthropogenic emissions. We believe that a comprehensive assessment of the climatic and economic impacts of several key nonlinear Arctic feedbacks warrants a high-impact publication for the reasons described earlier. Furthermore, the Reviewer is not correct in basing their judgement only on the GMST effect of the two feedbacks, which could indeed be deemed relatively small, especially during the current century. The paper clearly states that by considering the NPV of the additional economic impacts until 2300 alongside the GMST changes, we can see a bigger picture. The statistically significant mean increases in the total economic effect climate change due to the PCF and SIAF amount to 3.6% (\$18trillion) under the 1.5°C scenario, 7.6% (\$33trillion) under the 2°C scenario, 11.1% (\$109trillion) under levels of mitigation consistent with the current national pledges, and by 7.6% (\$123trillion) under the business as usual scenario, which is outlined in the Abstract and the Summary. This makes a compelling case for performing a similar assessment of the other known tipping elements and feedbacks.

An additional value of our results is the new nonlinear mechanistic understanding of the two feedbacks. For example, we show the importance of ocean uptake in neutralizing and even reversing the PCF impact, and the decline of the warming signal due to the SIAF, both of which occur in the 23rd century under the high emissions scenario. These and other results are based on the multiple layers of nonlinearity in the PCF and SIAF captured by the emulators for the first time, potentially leading to similar discoveries for the other essential planetary feedbacks.

- *“... leading to a growth of the permafrost area and refreezing of some of the thawed carbon”. The description of the PCF is not particularly clear in terms of a necessary distinction between thawed and emitted carbon. Does the sentence “This reduces the amount of thawed carbon, causing a decrease in the PCF strength” mean that thawed permafrost carbon is explicitly represented in the model, or is it conceptually the same as “emitted carbon”? Looking at the equations in the methods section, it seems that the process of carbon accumulation and emission are represented by the same equation, although the processes (and the timescales)*

are very different: carbon accumulation in permafrost areas (and anywhere else) is a very slow process, while carbon emission can be relatively fast (even if it still takes decades to centuries). Some strange features of the strong mitigation scenarios as calculated by the IAM presented here might be explained by this unrealistic symmetry in the model formulation and response.

First, if the “model” referred to by the Reviewer is SiBCASA, then the “thawed” and “emitted” permafrost carbon are two different things. However, the PCF emulator only models the emitted carbon explicitly, while also accounting for the time lags between the temperature rise, thawed carbon and emitted carbon. Although there is only one lag parameter, it is a nonlinear function of the difference between the equilibrium and actual cumulative emissions, which appears to account for the underlying sequence of events, starting with the warming and ending with the carbon being emitted. Achieving this level of accuracy, while retaining the relative simplicity, is a step further from the previously used linear PCF emulator (Kessler, 2017).

Second, we agree with the Reviewer that burial of carbon occurs over geologic time scales of tens of thousands of years. Therefore, we modified the PCF emulator based on assumption that none of the carbon emitted is refrozen into permafrost on the timescales considered. The 1.5°C and 2.0°C scenarios still show a decline of the GMST effect of the PCF. This is due to the drop in annual permafrost carbon emissions stemming from the global warming slowdown and the eventual cooling under these scenarios, accompanied by ocean carbon removal. In addition, some of the permafrost carbon is removed as part of negative CO₂ emissions that are central to these scenarios.

We modified the text to clarify all these points.

- “For the High fossil fuels use scenario...”: for clarity, you might add here that, in addition, the PCF is exhausted in this scenario after 2250.

The climate scenarios, the PCF emulator and the CO₂ cycle are all different in the revision, but the permafrost carbon stock exhaustion does occur for BaU in the 23rd century. We clarified this in the text.

- Figure 6: One wonders what the NPV for very low GMST changes is in the “default impact” scenario. Should probably be 0 but the figure (which invites to extrapolate by eye) does not suggest it.

The figure in question and the underlying modelling of the economic impacts of climate change have all been updated in the revision. The results presented in the revised manuscript are based on a new implementation of the Burke et al. (2015) economic impact function, the most comprehensive to date, within the modelling framework of the PAGE IAM. As in the original version of the results and the figure in question, the revision still presents the “total economic effect” of climate change (Figure 6, left panel), which includes mitigation and adaptation costs in addition to the climate-driven economic impacts from Burke et al. It is because of the progressively higher mitigation costs that the lowest emissions scenarios start

becoming more expensive in terms of the total economic effect. We illustrate this point by adding the imaginary Zero Emissions scenario (anthropogenic emissions stop immediately in 2015), which provides a hypothetical upper bound for the mitigation costs, while also leading to the lowest possible climate impacts due to the historic emissions only. The 1.5°C scenario appears to be marginally more expensive than the 2.0°C scenario for exactly the same reason, although we found that the two are not significantly different at the 5% and 10% significance level.

We clarified these points in the main text and in the caption to the Figure 6.

- *“These results do not include other known climatic feedbacks...”: The whole sentence is not understandable, at least not to me, probably as a result of a copy-paste that went wrong. It looks like two fragments of independent lines of thought.*

The text in question has been revised substantially. Our response to the next comment clarifies the line of reasoning adopted in the revision.

- *The two sentences following immediately after (“Science is still limited... additional losses from the two Arctic feedbacks.”) are also unclear. Why should an underestimate be likely? Why not an overestimate?*

This section has been revised substantially. PAGE-ICE model, in common with other IAMs, does not explicitly model other known climatic tipping elements such as Amazon rainforest, boreal forest, coral reefs and ENSO, as well as ocean acidification, climate-induced large-scale migration and conflict (Kriegler et al., 2009), although these effects are included implicitly in the highly uncertain “non-economic” and “discontinuity” impact sectors in PAGE-ICE. Based on the assessment of the effect of the nonlinear PCF and SIAF carried out here, it is possible that the economic effect of climate change could increase further if the other climatic and societal tipping elements are modelled explicitly. The same applies to the inclusion of a comprehensive representation of the impacts of rising temperatures and growing extreme weather events on economic growth (Dell et al., 2014), which is still missing in the literature. However, we do agree with the Reviewer that there is no justification in the results for making a stronger statement such as “an underestimate is likely”.

We made all these issues clear in the revised text.

- *“For most scenarios... PCF... impacts are greater for higher emissions scenarios.” Of course they are, but relative impacts are more interesting.*

We provide both absolute and relative estimates for the statistical means of the combined economic effect of the nonlinear PCF and SIAF in the Abstract and the Summary. The complete statistical results (5-95% range, quartiles, median, mean) are shown for the absolute quantities only for the sake of clarity (Figure 6).

- *The ECS in CMIP exercises is typically calculated using the abrupt4xCO₂ experiment (e.g. Andrews et al., GRL 2012, doi;10.1029/2012GL051607). At 4xCO₂, there is not much sea-ice left. This means that the ECS used by default should take this sea-ice depletion into account. This does not appear clearly enough in the paper (but it does not annihilate the progress made by taking into the variability of the SIAF, higher in the beginning than at the end of high emission scenarios).*

Unlike the CMIP5 definition of the ECS, which is based on the abrupt 4xCO₂ experiment, PAGE-ICE employs the more generic IPCC AR5 definition based on the 2xCO₂ increase relative to pre-industrial conditions. The 2xCO₂ ECS parameter in PAGE-ICE is therefore consistent with the range in IPCC AR5, which is based on pale-records, CMIP5 simulations and 2xCO₂ experiments in climate emulators of intermediate complexity. The corresponding mean equilibrium warming is 2.8°C (5-95% range of 1.7°C to 4.2°C), and our updated SIAF representation indicates that these levels of warming are still well below the temperatures when the SIAF starts to decline with the loss of the winter sea ice. For the GMST anomaly equal to the ECS, the SIAF averages at 0.18 W/m²/°C (5-95% range of 0.15 to 0.21 W/m²/°C), which takes into account the first peak in the SIAF associated with the loss of the late summer sea ice.

All these points are made clear in the text.

To provide further justification for our ECS parameterisation, we conducted the abrupt 4xCO₂ and transient 2xCO₂ and 4xCO₂ experiments in PAGE-ICE. The results are consistent with IPCC AR5 (Tables 9.5 and 9.6, IPCC AR5 WG1), and are summarised in Supplementary Materials.

Finally, we analysed the 4xCO₂ sea ice simulations in multiple CMIP5 models, and many of them still have the ice left in several months of the year towards the end of the 150-year experiments. They do not cover the range that the RCP8.5 scenario extended until 2300 does, although they go through similar nonlinear transitions in the SIAF initially.

- *The summary paragraph starting with “The climate policy implications of the Arctic feedbacks are profound.” comprises in a nutshell the main troubles I have with the paper. The points mentioned in the paragraph show are either that the results presented here are not new or show that these results do not induce a substantially renewed view of the problem. It is not new that the PCF, not taken into account in CMIP5 climate models (and thus in most IAMs) requires additional mitigation efforts. It is no new that a better understanding of the complex processes in the Earth’s climate system is required. The paper shows (again) that the PCF and the SIAF become less important at higher GMST changes, which is somewhat at odds with the statement in this paragraph that “triggering these planetary feedbacks might accelerate the pace of climate change”. And it is not clear to me why the results presented here should show why current IAMs “likely... underestimate the total economic effect of climate change”: the effect of the PCF (increased cost) has been shown by the authors before and the results here show reduced cost in the long term. They do not show any clear results concerning other tipping points that warrant the assessment that the economic costs of climate change will be higher than current assessments indicate.*

The Summary section has been revised substantially, and we avoid making the statements like those flagged by the Reviewer. This also applies to the paper as a whole. The key added value of our findings and their potential implications for science and policy are stated clearly and concisely, as is described in our earlier comments.

“Further interdisciplinary research is needed to improve our understanding of the complex nonlinear interactions both in the Earth’s climate and socio-economic systems.” IMHO, there is first a particular need for disciplinary research to improve the understanding of nonlinear interactions within the Earth’s climate and within the socio-economic systems separately.

We agree with the Reviewer that further disciplinary research is also needed, although we believe it should go hand in hand with interdisciplinary research. Both are equally crucial for addressing the complex challenges associated with climate change. We modified the statement accordingly to reflect on this.

Some remarks on the methods section: Please define “warming degree-years”

This parameter has been excluded from the analysis following a substantial revision of the methane component of the PCF emulator. The revised methane component has the same functional form as the CO₂ component, but the parameter ranges obtained from the statistical fitting to the SiBCASA simulations are very different between the two. This is described in detail in Supplementary Materials.

“The nonlinear Equation 1 and Equation 2 are solved analytically...” Really? Not numerically?

Following an established methodology for the PAGE IAM, we solve the equation in question in closed form on each finite analysis period $t_{i-1} < t < t_i$ of the PAGE model, which is possible since the temperature is assumed to be constant during each period. The resulting numerical scheme is described in Supplementary Materials. Using the closed form solutions with piece-wise constant temperatures (and also emissions) is the only way to make the model work for the long analysis periods (first 10, then 25 and finally 50 years) since many physical processes represented by the model take place on either similar or shorter timescales.

The parameterization of the SIAF does not necessarily appear robust with respect to the choice of the ESMs. It might be more reasonable to use a more simplified parameterization based on only two or three eras.

As mentioned earlier, we carried out extensive new analysis of the SIAF using a wider pool of CMIP5 models and including their historic simulations in addition to RCP8.5 projections, and remodelled the SIAF emulator to make it statistically robust. The new emulator has only three eras and is based on just four statistically independent uncertain parameters. The details are provided both in the main text and in the Methods sections.

The scientific literature clearly distinguishes between the Charney-type ECS and the Earth System ECS, which basically includes feedback strength variations as those discussed here. The authors might want to reflect this discussion here.

As we mentioned earlier, the ECS parameter in PAGE-ICE is with the range in IPCC AR5, which is based on paleo-records, CMIP5 simulations and 2xCO₂ experiments in climate emulators of intermediate complexity. This is described in the Methods section.

Reviewer #2 (Remarks to the Author)

This paper provides new and interesting insights on the climatic and economic implications of two important Arctic related process, i.e. the permafrost carbon feedback (PCF) and the sea-ice albedo feedback (SIAF). Recently, different studies have captured and explored the implications of these two elements in isolation and/or in exogenous way in IAM models. However, most of these studies have not disentangled, for example, the level of SIAF that is typically included in the equilibrium climate sensitivity (ECS) parameter. The novelty of this paper is that both elements are, for the first time, considered in combination and, also that both effects are endogenously and transparently captured within the model. This is done via coupling an emulator of both physical process within a well-known IAM model (PAGE model), which was used for the Stern Report.

We would like to thank the Reviewer for their appreciation of the main advances of our work. In the revision, we strive to put more emphasis of the “endogenous and transparent manner” of our modelling, and to communicate the key results and their policy implications clearly and concisely.

The calculations seem to have been done competently and the result is written up in a transparent way. The implications of this paper will be of the interest of the scientific community on climate change. Moreover, and given the rapid changes that we are seeing in the Arctic, I think that studies like this will be relevant for climate policy and for policy makers as it helps to improve our understanding on the future impacts from climate change and also of the risk associated to crossing some critical thresholds. Therefore, I think this work is acceptable for publication if the following questions are clarified:

We agree with the Reviewer’s statement that “given the rapid changes that we are seeing in the Arctic, I think that studies like this will be relevant for climate policy and for policy makers...” We strive to bring these points out strongly in the revision.

1) *Scenario zero emissions. Add another scenario where GHG emissions are immediately reduce to zero forever. Although this scenario is not realistic it would give to this paper an interesting new insight which is to know the increase in the temperature we can expect due to the inertia of the climate system with and without the PCF and SIAF. This scenario will also*

be very useful in order to compare the result obtained in this paper with the PAGE-ICE model with those from the IPCC AR5 report (Figure 1, FAQ 12.3, Chapter 12, WG1).

We implemented the Zero Emissions scenario following the helpful suggestion by the Reviewer. This scenario is a useful indicator both in terms of estimating residual impacts of the historic emissions, as the Reviewer pointed out, but also and in terms of providing a hypothetical upper bound for the mitigation costs. We reflected on these aspects in the text.

Comparing with the Figure 1, FAQ 12.3, Chapter 12, IPCC AR5 WG1, our results for the GMST projections (Figure 3) under the Zero Emissions scenario do not show the near-immediate spike (less than 1 decade) in the GMST effect due to the disconnection of the cooling from sulphate aerosols. The reason for this is that PAGE-ICE RF from all GHGs, averaged over the surface of the planet, shows a marginal drop immediately after all the emissions are set to zero, which is not unreasonable given the dominance of the RF from the long-lived CO₂ clearly documented in IPCC AR5.

What is more important for the purposes of our study is that the century-long GMST projections of PAGE-ICE across all the scenarios, including Zero Emissions, are consistent with the literature. To improve the accuracy of the long-term projections, we remodelled the CO₂ cycle in the version of PAGE-ICE used for the revised manuscript, based on the latest multi-model assessment of the atmospheric CO₂ response function by Joos et al. (2013). To provide justification for our parameterisation of the GMST response to the RF, including the characteristic time lag associated with the ocean heat uptake, we conducted the abrupt 4xCO₂ and transient 2xCO₂ and 4xCO₂ experiments in PAGE-ICE. The results are consistent with IPCC AR5 (Tables 9.5 and 9.6, IPCC AR5 WG1), and are summarised in Supplementary Materials.

2) Scenario NDCs no-US. Although I like the structure of the scenario, the “NDCs no-US” scenario is bit out of scope when doing projections to 2100 and 2300. The scenario can be maintained but I would recommend not to highlight it in the abstract and better to concentrate more the attention on the 1.5 and 2-degree scenarios which are the policy relevant. Finally, notice that after the COP 21 and the Paris Agreement the contributions are not anymore “intended”.

We renamed this scenario as “NDCs Partial” and removed it from the Abstract, concentrating on the more policy-relevant 1.5°C and 2°C scenarios, a scenario with levels of mitigation consistent with NDCs, and business as usual (BaU) scenario. We also added a 2.5°C target scenario elsewhere in the paper, which is more ambitious than the NDCs but falls short of the 2°C target. The NDCs Partial pathway is consistent with an estimated long-term effect of the US’s withdrawal from the NDCs, should the current federal policy persist. Of course, we acknowledge that the US could re-commit to the NDCs on the federal level during the forthcoming political cycles, while some of the other big emitters currently pledging to cut the emissions may fail to do so, or withdraw from the deal altogether. Thus, the NDCs Partial pathway is still relevant as a means of describing a possible future with only partially implemented NDCs.

3) Models selected for emulator calibration. It is not clear (not for PCF nor for the SIAF) which is the criteria used for selecting some of CMIP5 models versus others. This is crucial for the calibration of the emulators and, therefore, for the results of the paper. Please, explain briefly in the manuscript even if a more technical detail is needed or included in the SM.

For the PCF, we chose the 5 models that come with extended RCP8.5 and RCP4.5 runs out to 2300, and provide a plausible range of climate projections consistent with that of the entire CMIP5 models' set. For the SIAF, we extended the pool of models used by choosing all the CMIP5 models that have runs from 1850 until 2300 under the RCP8.5 scenario, with exception of GISS-E2-H, GISS-E2-R and IPSL-CM5A-LR. These models produce unrealistically high extents of Arctic sea ice in comparison with the satellite-derived sea ice concentration (Riemann-Campe et al., 2014), and/or are outliers in terms of the projected sea ice extent relative to the CMIP5 mean. We also added results from GFDL-CM3, MIROC-ESM and NorESM1-ME run between 1850 and 2100 under RCP8.5. Together with MPI-ESM-LR and CCSM4, these models achieve the lowest misfit with historic satellite sea ice records across the Arctic Ocean, and help calibrate the nonlinear transitions in the SIAF up until around 5°C of warming relative to pre-industrial conditions, which is the range most relevant for climate policy.

We provided clear explanations in the text. We also added two tables to the Supplementary Materials that summarise the models used for the SIAF calibration, list the key criteria for choosing or not choosing them, and provide references to the model descriptions.

4) Uncertainty analysis. The paper results are based on a Monte-Carlo generated probability distribution of "multiple climatic and economic parameters". Please, explain which are those parameters and what are the distribution selected for them.

We listed all the 162 uncertain parameters used in PAGE-ICE in Supplementary Materials, and referred to this in the main text as well as in the Methods section.

5) SIAF process. The figure 2 of the paper is not explained. Also, I am not convinced about the step-approximation followed (figure 8), why eight distinct eras? which is the foundation for this, is statistically-related or ice-process-related or both? It is not clear. Also, it is quite strange that the MPI-ESM-LR model match/fit 100% to this, how this could be possible?

Following from the Reviewers' feedback, we carried out extensive new analysis of the SIAF using a wider pool of nine CMIP5 models that were deemed fit for the purpose, including their historic simulations in addition to RCP8.5 projections, and remodelled the SIAF emulator to make it statistically robust. The new emulator has only three eras and is based on just 4 statistically independent uncertain parameters derived from the multi-model SIAF. We do not consider peaks and troughs in the SIAF separately for each model. The details are provided both in the main text and in the Methods section.

The accuracy of each model's fit to historic sea ice data from satellite records is characterised by a standard root mean squared misfit parameter. The misfit has the lowest value for the

MPI-ESM-LR model, which is used as a benchmark and is assigned the highest ranking of 1, while the misfits for all the other CMIP5 models are higher in comparison. The respective rankings for the other CMIP5 models are defined as the inverses of their misfits scaled with the MPI-ESM-LR misfit, and are smaller than 1 as a result.

6) Cumulative RF: Please, explain the concept of “cumulative” radiative forcing of Figure 5 and how do you calculate this.

We removed the word “cumulative” from the term “RF” in association with the SIAF, opting to use just “RF” instead to avoid any confusion. The RF from the additional solar absorption as a result of the sea ice loss is calculated as the integral of the SAIF with respect to temperature over a given GMST increase. This definition is clear from the main text and the formulas in the Methods section.

7) SIAF and winter sea ice. You say that “constant SIAF greatly overestimates the GMST increases compared to the non-linear SIAF beyond 2150, which is due to the loss of the winter sea ice”. The role of winter sea ice is not totally clear in this sentence. If I understand this correctly, the issue is that there is a saturation point (when winter sea ice is melt) where the extra warming cannot increase more (there is no sea ice anymore) and that constant SIAF does not capture this maximum level, right?

We revised the text of the section in question considerably following the remodelling of the SIAF emulator. Instead of referring to the “loss of winter sea ice”, we say “year-round loss of the sea ice” and “complete disappearance of the sea ice”. The Reviewer is correct in their interpretation of the “saturation point” beyond which the nonlinear SIAF drops to zero and the corresponding RF does not increase further. A constant SAIF implies that the RF continues to increase linearly with the temperature, leading to an overestimation of the warming effect associated with the sea ice loss.

8) Data of Table 1. Please check that the all data of this table is correct and also that data reference in the text match with the table. For example, the total economic impact of the 1.5 scenario (\$566tr) is higher than the 2°C scenario (\$476tr), is this correct? Are the extra-damages in PAGE model greater than the extra-mitigation cost of going from 2°C to 1.5°C? I guess that Burke’s estimates would change this.

The results for the economic effect of climate change and of the nonlinear Arctic feedbacks presented in the revised manuscript are based on a new implementation of the Burke et al. (2015) impact function. We adapted it to fit with the single year consumption-only approach for climate impacts in PAGE, which provides an incremental change in the modelling framework. We also revised autonomous learning rates for the mitigation costs and updated the sea level rise calibration. The new results for the economic impacts are given with full uncertainty ranges (5th, 25th, 75th, 95th percentiles, median and mean) in the new Figure 6, while the Table 1 has been removed.

The 1.5°C scenario still appears to be marginally more expensive than the 2.0°C scenario (statistical mean values of \$505 trillion vs \$438 trillion), although we found that the two are not significantly different at the 5% or 10% significance levels. As we do not explicitly model other climatic and societal tipping elements, possible co-benefits of deep mitigation, and impacts of rising temperatures and growing extreme weather events on economic growth, these factors could further reduce the mean economic effect of the 1.5°C scenario relative to the 2.0°C scenario.

We clarified all these points in the main text and in the caption to the Figure 6.

9) Coupling effect: Also explain better in the text when you refer in Table 1 to the “coupling effect”, it is not clear. The coupling effect is to me very relevant and the results of the table could also be used to connect with the conclusion and further research section when you mention that your results are likely an underestimation as some processes are not captured. Although the SIAF effect seems small compared to the PCF, I wonder if due to its coupling effect with others major systems (Greenland, THC, etc.) this effect could be still critical.

After additional considerations, we concluded that the interaction between the PCF and SIAF themselves is of a lesser importance due to its relatively small magnitude, and therefore we do not focus on it in the revision. Instead, we analyse statistical significance of the impacts of the nonlinear PCF and SIAF, separately and combined, on global climate and economy, and explain why focusing on the nonlinearities in these two high latitude feedbacks is relevant for climate policy, despite the multiple underlying uncertainties in the climate such as the cloud feedback.

The methodology introduced in this paper could be used to quantitatively assess climate policy implications of other tipping elements in the Earth’s climate system such as Greenland and West Antarctic ice sheets, Amazon rainforest, Boreal forest, Sahel and ENSO. As such, we believe this work has clear potential to influence both climate science and policy.

Reviewer #3 (Remarks to the Author)

This paper looks at the economic impact of including both the permafrost carbon feedback and the sea ice albedo feedback in an integrated assessment model (IAM). The work builds on the paper by Hope and Schaefer (2016) which looks at the financial consequences of including just permafrost CO₂ and CH₄ release in an IAM. It includes more scenarios (particularly the 1.5 and 2.0 degree stabilisation scenarios) but a relatively limited set of uncertainties. The sea ice albedo feedback has previously been included in an IAM but in a simpler form.

We do not agree with the Reviewer that the work is based on a relatively limited set of uncertainties.

First, the PAGE-ICE model has over 150 uncertain parameters calibrated according to the latest insights from climate science and economics. These parameters are listed in the Supplementary Materials.

Second, both the PCF and SIAF emulators represent a plausible range of uncertainties in these feedbacks. For the PCF emulator we used SiBCASA simulations with climates from five CMIP5 models run under two contrasting emissions scenarios out to 2300, which provides a plausible range of climate projections consistent with that of the entire CMIP5 set. For the SIAF emulator, we carried out extensive new analysis of the SIAF as part of the revision using sea ice extent estimates from a wider pool of nine CMIP5 models. We chose the models that provide a good match with satellite records for Arctic sea ice and do not produce outliers in sea ice projections under RCP8.5 out to 2300.

There are some relatively small interactions between the permafrost feedback and the sea ice feedback.

After additional considerations, we concluded that the interaction between the PCF and SIAF themselves is of a lesser importance due to its relatively small magnitude, and therefore we do not focus on it in the revision. Instead, we analyse statistical significance of the impacts of the nonlinear PCF and SIAF, separately and combined, on global climate and economy, and explain why focusing on the nonlinearities in these two high latitude feedbacks is relevant for climate policy.

In the abstract they pick out one key result regarding the cost of the two Arctic feedbacks in 2300 of the USA withdrawal from the Paris agreement. Studies of the permafrost carbon feedback typically go out beyond 2100 because at the longer time scales the carbon has had more time to decompose and be released and have a notable impact. However, in the abstract I might also pick out another key result for the lower temperature targets.

We renamed this scenario as “NDCs Partial” and removed it from the Abstract, concentrating on the more policy-relevant 1.5°C and 2°C scenarios, a scenario with levels of mitigation consistent with NDCs, and business as usual scenario.

Results from SiB-CASA are used to calibrate the permafrost carbon model. The setup of SiB-Casa was originally described by a Schaefer et al. (2011) paper - further details are still needed as to how the methane release from the permafrost is estimated.

As permafrost thaws, aerobic and anaerobic microbial decomposition of organic carbon leads to the release of methane and CO₂ (Schuur et al., 2008), which SiBCASA accounts for in calculating the carbon flux. We reflected on this in the text.

The estimate of permafrost carbon lost as CO₂ and CH₄ is high compared with previous work (for example Burke et al. 2017; Schneider von Deimling et al., 2015) Whilst this is not necessarily wrong, it probably needs to be discussed.

The range of CO₂ and methane release from permafrost can vary between different LSMs, and it can be higher in SiBCASA than in several other models. However, using the SiBCASA output for the PCF emulator produces reasonable estimates of carbon fluxes that fall roughly in the middle of available estimates from other LSMs, and thus is a justified representative for the range of the available models (Schaefer et al., 2014; Schuur et al., 2015; McGuire et al., 2018). These estimates suggest that total carbon release by 2100 from thawing permafrost is 120 ± 85 GtC for the RCP8.5 scenario, while the corresponding carbon release modelled by SiBCASA is 104 ± 37 GtC (Schaefer et al., 2011), well within the expected range seen in the published literature.

If all of the permafrost carbon is thawed by 2300 the majority of it will have to have decomposed to emit 500 Gt C by 2300. This decomposition process is relatively slow if it is just related to active layer deepening. Therefore, I think these high losses need to be justified more carefully.

In our opinion, it is mainly the physical process of permafrost thaw (thermodynamics) that is slow. Once the frozen organic carbon is thawed, microbial activities will immediately start decomposing and releasing the old carbon as methane and CO₂. The nonlinear PCF emulator only models the emitted carbon explicitly, while also accounting for the time lags between the temperature rise, thawed carbon and emitted carbon. Although there is only one lag parameter, it is a nonlinear function of the difference between the equilibrium and actual cumulative emissions, which appears to account for the underlying sequence of events, starting with the warming and ending with the carbon being emitted.

In addition, the temperature responses to the permafrost carbon feedback in Figure 4a are different to those found in Burke et al., 2017, Figure 7. Is this caused by the additional inclusion of methane?

The higher temperature responses are partially due to the lagged CO₂ ocean uptake not represented by the IMOGEN climate emulator used in the Burke et al. (2017a,b) studies, which assumes that the ocean uptake happens instantaneously (Mauritsen & Pincus, 2017). The addition of the methane component (not included in the Burke et al. work) does increase the temperature further. Finally, the characteristic timescales of the permafrost active layer deepening and organic carbon decomposition could be different between SiBCASA and the JULES and ORCHIDEE-MICT LSMs used in the Burke et al. studies.

We reflected on these points in the text.

I feel the uncertainties in the Arctic feedbacks need to be significantly expanded on. There are many uncertainties in the release of permafrost carbon which are not even discussed. Also the uncertainties need to feed through into the economic impact.

As long as the uncertainties are captured by the underlying process-based biophysical models, they will feed into the Arctic feedback emulators, subsequently affecting the simulated coupled climate and the resulting economic impacts in PAGE-ICE.

To address the issue of uncertainties in the Arctic feedbacks, we carried out extensive new analysis of the SIAF using a wider pool of CMIP5 models and including their historic simulations in addition to RCP8.5 projections, and remodelled the SIAF emulator to make it statistically robust. We also remodelled the PCF emulator by adding further layers of nonlinearity to the permafrost carbon emissions in response to GMST rises, while also achieving a more accurate and statistically robust fitting to the SiBCASA results. The revised emulator has a more generic form in order to make it suitable for reproducing simulations from other LSMs, which provides a worthy topic for a follow-up study.

Finally, the dominant source of uncertainty in the PCF is the uncertainty in the projected climate ($\pm 34\%$). The second largest source of uncertainty is the amount of carbon in permafrost ($\pm 15\%$) (Schaefer et al., 2011). We agree that there are other sources of uncertainty, but they do not significantly influence the overall uncertainty in the estimated PCF.

All these points are discussed in the main text, Methods section and Supplementary Materials.

The most relevant timescales of these economic impacts need to be explained more carefully. There is both an instantaneous impact and a longer term impact. This could be made clearer and is definitely of interest.

This is a very good point. One needs to distinguish between climatic and economic timescales. On the climate side, the radiative forcing effect is instantaneous for the SIAF and is delayed for the PCF (through lags in permafrost carbon thawing, decomposition and ocean carbon uptake). There is also a lag between the radiative forcing and GMST associated with the ocean heat uptake.

On the economy side, there is an ongoing debate about the long-term nature of climate-driven impacts on economy. The literature distinguishes between the level effects (instantaneous but with no long-term consequences) and growth effects (persistent) of climate on economy. In the revised manuscript we used the macro-econometric analysis of historic temperature shocks on economic growth in multiple countries by Burke et al. (2015), the most comprehensive of its kind to date. However, we adapted it to fit with the level effects' approach to climate impacts in the PAGE model, which provides an incremental change in the modelling framework.

We clarified these points in the text.

Economics - A "discounted economic impact" is a little confusing term.

Discounting of cash flows that occur at different times is a fundamental concept in economics. It is captured by the 'Net Present Value' calculation, which is commonly used to present the results of longer-term cost-benefit analyses, including that for climate change. We believe that our description of the economic calculations is sufficient and no further clarifications are necessary.

References

Winton, M. (2006). Does the Arctic sea ice have a tipping point?. *Geophysical Research Letters*, 33(23).

Winton, M. (2008). Sea ice–albedo feedback and nonlinear Arctic climate change. *Arctic sea ice decline: Observations, projections, mechanisms, and implications*, 111-131.

Hope, C., & Schaefer, K. (2016). Economic impacts of carbon dioxide and methane released from thawing permafrost. *Nature Climate Change*, 6(1), 56-59.

González-Eguino, M., & Neumann, M. B. (2016). Significant implications of permafrost thawing for climate change control. *Climatic Change*, 136(2), 381-388.

Kessler (2017) Estimating the economic impact of the permafrost carbon feedback, *Clim. Change Econ.*, 08, 1750008. <https://doi.org/10.1142/S2010007817500087>.

González-Eguino, M., Neumann, M. B., Arto, I., Capellán-Perez, I., & Faria, S. H. (2017). Mitigation implications of an ice-free summer in the Arctic Ocean. *Earth's Future*, 5(1), 59-66.

Burke, E. J., Ekici, A., Huang, Y., Chadburn, S. E., Huntingford, C., Ciais, P., ... & Krinner, G. (2017). Quantifying uncertainties of permafrost carbon–climate feedbacks. *Biogeosciences*, 14(12), 3051.

Burke, E. J., Chadburn, S. E., Huntingford, C., & Jones, C. D. (2018). CO₂ loss by permafrost thawing implies additional emissions reductions to limit warming to 1.5 or 2° C. *Environmental Research Letters*, 13(2), 024024.

Lenton, T. M., Held, H., Kriegler, E., Hall, J. W., Lucht, W., Rahmstorf, S., & Schellnhuber, H. J. (2008). Tipping elements in the Earth's climate system. *Proceedings of the national Academy of Sciences*, 105(6), 1786-1793.

Schellnhuber, H. J., Rahmstorf, S., & Winkelmann, R. (2016). Why the right climate target was agreed in Paris. *Nature Climate Change*, 6(7), 649-653.

Riahi, K., Van Vuuren, D. P., Kriegler, E., Edmonds, J., O'neill, B. C., Fujimori, S., ... & Lutz, W. (2017). The shared socioeconomic pathways and their energy, land use, and greenhouse gas emissions implications: an overview. *Global Environmental Change*, 42, 153-168.

Christensen, P., Gillingham, K., & Nordhaus, W. (2018). Uncertainty in forecasts of long-run economic growth. *Proceedings of the National Academy of Sciences*, 115(21), 5409-5414.

Yumashev, D. (2018). Why scientists have modelled climate change right up to the year 2300? <http://theconversation.com/why-scientists-have-modelled-climate-change-right-up-to-the-year-2300-92236?>

Anthoff, D., Hepburn, C. and Tol. R.S.J. (2009). Equity weighting and the marginal damage costs of climate change, *Ecological Economics*, Volume 68, Issue 3, 836-849

Hope, C. (2015). The \$10 trillion value of better information about the transient climate response. *Phil. Trans. R. Soc. A*, 373(2054), 20140429.

Cox, P. M., Huntingford, C., & Williamson, M. S. (2018). Emergent constraint on equilibrium climate sensitivity from global temperature variability. *Nature*, 553(7688), 319.

Brown, P. and Caldeira, K. (2018) Greater future global warming inferred from Earth's recent energy budget, *Nature*, 552,45–50, doi:10.1038/nature24672.

Huntzinger, D. N., Schwalm, C., Michalak, A. M., Schaefer, K., King, A. W., Wei, Y., ... & Berthier, G. (2013). The North American carbon program multi-scale synthesis and terrestrial model intercomparison project—Part 1: Overview and experimental design. *Geoscientific Model Development*, 6(6), 2121-2133.

Schaefer, K., T. Zhang, L. Bruhwiler, and A. P. Barrett (2011), Amount and timing of permafrost carbon release in response to climate warming, *Tellus Series B: Chemical and Physical Meteorology*, 63(2), pg. 165-180, DOI: 10.1111/j.1600-0889.2011.00527.x.

Schaefer, K, H Lantuit, VE Romanovsky, EAG Schuur, and R Witt (2014). The impact of the permafrost carbon feedback on global climate, *Env. Res. Lett.*, 9, 085003 (9pp) doi:10.1088/1748-9326/9/8/085003

Schuur, E. A. G., McGuire, A. D., Schädel, C., Grosse, G., Harden, J. W., Hayes, D. J., ... & Natali, S. M. (2015). Climate change and the permafrost carbon feedback. *Nature*, 520(7546), 171-179.

McGuire, AD; Lawrence, DM; Koven, C; Klein, JS; Burke, E; Chen, GS; Jafarov, E; MacDougall, AH; Marchenko, S; Nicolsky, D; Peng, SS; Rinke, A; Ciais, P; Gouttevin, I; Hayes, DJ; Ji, DY; Krinner, G; Moore, JC; Romanovsky, V; Schaedel, C; Schafer, K; Schuur, EAG; Zhuang, QL (2018), Dependence of the evolution of carbon dynamics in the northern permafrost region on the trajectory of climate change, *Proc Natl Acad Sci*, 115/15, p. 3882-3887, DOI: 10.1073/pnas.1719903115

Riemann-Campe, K., Karcher, M., Kauker, F. & Gerdes, R. (2014). Results of Arctic ocean-sea ice downscaling runs validated and documented. FP7 Project ACCESS D1.51 report. European Commission. Url: <http://www.access-eu.org/modules/resources/download/access/Deliverables/D1-51-AWI-final.pdf> .

Kriegler E, Hall JW, Held H, Dawson R, Schellnhuber HJ. (2009). Imprecise probability assessment of tipping points in the climate system. *Proc. Natl. Acad. Sci. U. S. A.* **106**: 5041–5046.

Dell, M., Jones, B. F., & Olken, B. A. (2014). What do we learn from the weather? The new climate-economy literature. *Journal of Economic Literature*, 52(3), 740-98.

Burke, M., Hsiang, S. M., & Miguel, E. (2015). Global non-linear effect of temperature on economic production. *Nature*, 527(7577), 235-239.

Mauritsen, T., & Pincus, R. (2017). Committed warming inferred from observations. *Nature Climate Change*, 7(9), 652.

Reviewers' comments:

Reviewer #2 (Remarks to the Author):

I am fully satisfied with the responses given to the 10 queries i raised and I also appreciate the effort done by the authors with the new analysis (zero emission scenario).

As I said in my first referee report I think this is timely paper that contributes to a relevant line of research. Therefore, I recommend its publication in its current form.

Yours faithfully

The reviewer

Reviewer #4 (Remarks to the Author):

The submitted manuscript presents an estimate of the additional climatic and financial impacts of both the permafrost carbon feedback and the sea-ice albedo feedback in the course of global climate change until the year 2300. To this end, the authors use an integrated assessment model which includes these two feedbacks in form of non-linear functions which are derived from more complex, process-based models. I was asked to specifically comment on the uncertainty analysis of these non-linear functions performed by the authors, considering the comments by reviewer #3.

Same as reviewer #3, I think that the uncertainty analysis regarding the permafrost carbon feedback is rather limited, for the following reason:

While SiBCASA may reproduce the average behaviour of several different land surface models (LSMs), using only one LSM completely neglects the uncertainty range resulting from different LSM parametrisations. The representation of permafrost carbon dynamics in LSMs is still under development. Cryoturbation, for instance, is generally missing in these models.

Consequently, different LSMs are expected to predict different dynamics of permafrost carbon for the same climate forcing in the next 300 years.

The authors address this point, which is also brought up by reviewer #1, by stating that uncertainty in climatic forcing is the most relevant factor for the total uncertainty of the permafrost carbon feedback (Schaefer et al, 2011). For this reason, they use 5 different CMIP5 models to generate an uncertainty range regarding the influence of climate on permafrost carbon dynamics. However, the newer study by Burke et al (2017), also cited by the authors, finds that "Structural differences between the land surface models (particularly the representation of the soil carbon decomposition) are found to be a larger source of uncertainties than differences in the climate response."

I therefore suggest to use at least the two LSMs from the study by Burke et al, 2017 (JULES and ORCHIDEE-MICT) to test if they produce similar permafrost carbon dynamics than SiBCASA.

If this is the case, the statement on the larger relevance of climate forcing for uncertainty of permafrost carbon release compared to LSM parametrisation would be better justified.

Otherwise, the uncertainty range from different LSMs should be incorporated into the total uncertainty of the estimates.

Furthermore, I would like to see a sensitivity of the total estimates to the value of the initial permafrost soil carbon stock (560 Gt). This estimate also comes with an uncertainty range which can be incorporated into the presented analysis.

Reviewer #5 (Remarks to the Author):

In the manuscript entitled, "Climate policy implications of nonlinear decline of Arctic land permafrost and sea ice," the authors use dynamic emulators of complex physical models in the integrated assessment model PAGE-ICE to estimate the impacts of the nonlinear feedbacks on the global climate and economy under a range of climate scenarios consistent with the Paris Agreement. Largely, the revisions resolved the previous comments. However, there are a few lingering concerns.

The authors focus on two feedbacks in the Earth's climate system, the permafrost carbon feedback (PCF) and the sea-ice albedo feedback (SIAF), which could accelerate global warming and cause additional economic impacts. As these feedbacks represent three of the thirteen "tipping elements" identified in another recent study, the implication made by the authors is that actual impacts could be substantially more severe. However, these two feedbacks do not represent a complete picture of the feedbacks that constrain equilibrium climate sensitivity. More importantly, since radiative feedbacks (water vapor, clouds, sea ice, etc.) interact with each other, it is not well posed to suppose that the SIAF is the only one acting nonlinearly, or that nonlinear behavior in the SIAF will not affect the other feedbacks. For instance, Caldwell et al. (2016) show significant covariances between albedo and shortwave cloud feedbacks, the latter which are negative in the Arctic. The consideration of the state-dependency of only a subset of high-latitude climate feedbacks is a limitation of the study and places a large caveat on the findings in regards to economic impacts.

As a minor aside, I was surprised by the authors' comment in their rebuttal that the snow albedo feedback is constant. Physically, I would expect it to operate under the same principle as the SIAF, i.e., the feedback approaches zero once no further melting can occur (as in Jonko et al., 2013). In the additional analyses the authors refer to, did they look at snow extent and/or snow albedo model output directly? I did not find the temperature amplification factor across the permafrost region to be a convincing proxy.

My second main comment pertains to the selection of CMIP5 models, and the meaning of uncertainty. The PCF emulator is driven by output from a different set of models than the SIAF emulator. The model mismatch compromises the physical consistency in the climate responses used as input to the IAM. I don't see how this would provide an objective, physically based measure of uncertainty in climate or economic impacts. I am also troubled by the heavy constraints used as model selection criteria for the SIAF. Firstly, the models are required to match satellite-derived sea ice concentration and secondly to match the ensemble-mean projected sea ice extent. It is not obvious that fidelity to a short and imperfect observational record is sufficient to capture correctly the climate response to anthropogenic forcing (e.g., Sanderson and Knutti 2012, Boe and Terray 2015). In addition, this is a very different definition of uncertainty than the one used for the PCF, which was to "provide a plausible range of climate projections consistent with that of the entire CMIP5 models' set." I think the results are potentially misleading, especially where the effects of both feedbacks are combined, given the model selection process.

References:

Caldwell, P.M., M.D. Zelinka, K.E. Taylor, and K. Marvel, 2016: Quantifying the Sources of Intermodel Spread in Equilibrium Climate Sensitivity. *J. Climate*, 29, 513–524, <https://doi.org/10.1175/JCLI-D-15-0352.1>

Jonko, A.K., K.M. Shell, B.M. Sanderson, and G. Danabasoglu, 2013: Climate Feedbacks in CCSM3 under Changing CO₂ Forcing. Part II: Variation of Climate Feedbacks and Sensitivity with Forcing. *J. Climate*, 26, 2784–2795, <https://doi.org/10.1175/JCLI-D-12-00479.1>

Boé, J. & Terray, L. *Clim Dyn* (2015) 45: 1913. <https://doi.org/10.1007/s00382-014-2445-5>

Sanderson, B. M., and R. Knutti (2012), On the interpretation of constrained climate model ensembles, *Geophys. Res. Lett.*, 39, L16708, doi: 10.1029/2012GL052665.

Response to Referees

We are grateful to the two replacement Reviewers for stepping in and providing a number of additional useful comments on our manuscript. We welcome the opportunity to further revise and resubmit the manuscript.

As during the first revision, to address the critical points raised by the Reviewers, we undertook substantial revisions of several key components of the manuscript. In particular, our focus has been on making the following major improvements to the modelling setup:

1. Including multiple land surface models (LSMs) in the estimates of the land permafrost emissions to represent the structural uncertainty in these models
2. Explicitly representing the land snow component of the surface albedo feedback, in addition to the sea ice component
3. Accounting for the changes in the Arctic cloud cover in response to variations in the surface albedo associated with the loss of the sea ice and land snow, which is known to be reducing the high-latitude surface albedo feedback in the 2xCO₂ equilibrium climate sensitivity (ECS) experiments in CMIP5 climate models
4. Harmonising the pools of climate models used to evaluate both the permafrost and albedo feedbacks and using a consistent model weighting scheme in order to eliminate the biases

Here we respond to the specific points raised by the Reviewers, listed in black italic. Our responses cover both the four major points listed above and a small number of additional minor issues. They form the basis for the revised manuscript presented here.

Addressing the four major points above led to the following changes in the results:

1. Using a combination of contrasting LSMs instead of the single LSM (SiBCASA) reduced the overall strength of the permafrost carbon feedback (PCF)
2. Adding the land snow albedo effect to the mix gave a stronger reduction in the strength of the surface albedo feedback (SAF) with temperature as the sea ice and land snow covers disappear
3. Accounting for the interactions between the surface albedo and the clouds in high latitudes reduced the strength of the SAF marginally

4. Using similar pools of climate models and applying a consistent weighting scheme based on model democracy throughout the study removed the heavy constraints that were previously imposed on the sea ice component of the SAF. While preserving the basic qualitative features of the nonlinear sea ice component, this update resulted in a more pronounced decline in the strength of the global SAF with temperature

With these crucial changes, we found that the permafrost carbon feedback (PCF) is still increasingly positive in warmer climates, while the surface albedo feedback (SAF) is either similar to or weaker than the legacy values that had been used in previous studies. The combination of these two factors still leads to sizeable increases in the mean discounted economic effect of climate change under the 1.5°C scenario, 2°C scenario and under the current national pledges (NDCs) from the Paris Agreement when compared with the legacy estimates.

Even though some of the key results changed from the previous version, the overall narrative of the relative importance of the two nonlinear Arctic feedbacks for climate policy assessments has been preserved.

Reviewers' comments:

Reviewer #4 (Remarks to the Author):

The submitted manuscript presents an estimate of the additional climatic and financial impacts of both the permafrost carbon feedback and the sea-ice albedo feedback in the course of global climate change until the year 2300. To this end, the authors use an integrated assessment model which includes these two feedbacks in form of non-linear functions which are derived from more complex, process-based models. I was asked to specifically comment on the uncertainty analysis of these non-linear functions performed by the authors, considering the comments by reviewer #3.

Same as reviewer #3, I think that the uncertainty analysis regarding the permafrost carbon feedback is rather limited, for the following reason: While SiBCASA may reproduce the average behaviour of several different land surface models (LSMs), using only one LSM completely neglects the uncertainty range resulting from different LSM parametrisations. The representation of permafrost carbon dynamics in LSMs is still under development. Cryoturbation, for instance, is generally missing in these models. Consequently, different LSMs are expected to predict different dynamics of permafrost carbon for the same climate forcing in the next 300 years. The authors address this point, which is also brought up by reviewer #1, by stating that uncertainty in climatic forcing is the most relevant factor for the total

uncertainty of the permafrost carbon feedback (Schaefer et al, 2011). For this reason, they use 5 different CMIP5 models to generate an uncertainty range regarding the influence of climate on permafrost carbon dynamics. However, the newer study by Burke et al (2017), also cited by the authors, finds that "Structural differences between the land surface models (particularly the representation of the soil carbon decomposition) are found to be a larger source of uncertainties than differences in the climate response."

I therefore suggest to use at least the two LSMs from the study by Burke et al, 2017 (JULES and ORCHIDEE-MICT) to test if they produce similar permafrost carbon dynamics than SiBCASA. If this is the case, the statement on the larger relevance of climate forcing for uncertainty of permafrost carbon release compared to LSM parametrisation would be better justified. Otherwise, the uncertainty range from different LSMs should be incorporated into the total uncertainty of the estimates.

We thank the Reviewer for these comments and acknowledge that we underappreciated the extent of variations between different LSMs, and their implications on the permafrost carbon emissions, in the previous versions of the manuscript. To rectify this issue, we invited the lead author of the Burke et al (2017) study to join the team and provide permafrost simulations of other suitable permafrost-enabled LSMs.

After careful considerations, we identified SiBCASA and JULES as two contrasting LSMs that are able to trace the fate frozen permafrost carbon under a specified future climate scenario, and therefore are suitable for calibrating the permafrost carbon feedback (PCF) emulator in the PAGE-ICE model. These two LSMs adopt different modelling philosophies to account for a number of highly uncertain biophysical processes in thawing permafrost soils. Compared to previously published values, JULES appears to be on the lower end and SiBCASA on the upper end of the reference multi-model studies, and the two models' responses to the warming have contrasting dynamics (see Figure 1 in Supplementary Materials and comments therein).

Based on these considerations, we argue that the combined use of SiBCASA and JULES in the revised manuscript provides a suitable estimate of the range of permafrost responses arising from uncertainty in LSM parameterisations. The simulations of ORCHIDEE-MICT and other LSMs include other carbon fluxes such as those associated with new vegetation growth in the areas with thawed permafrost, which makes them inapplicable for the purposes of our study.

We calibrated the PCF emulator separately for the SiBCASA and JULES simulations, with each LSM forced by multiple climate models under contrasting climate scenarios. The SiBCASA and JULES emulators were subsequently averaged with equal weights before linking with the carbon cycle in the PAGE-ICE model.

Furthermore, I would like to see a sensitivity of the total estimates to the value of the initial permafrost soil carbon stock (560 Gt). This estimate also comes with an uncertainty range which can be incorporated into the presented analysis.

To account for the uncertainty in the initial stock of old permafrost carbon, we scaled the permafrost emissions generated by the SiBCASA and JULES emulators according to the observed uncertainty range from Hugelius et al (2014). This was done directly within the PAGE-ICE model before linking the permafrost carbon emissions with the carbon cycle.

As a result, our representation of the PCF in PAGE-ICE includes each of the three crucial components of uncertainty:

- uncertainty in LSM parameterisations (represented by SiBCASA and JULES)
- uncertainty in climatic forcing (represented by multiple climate models and scenarios in the SiBCASA and JULES simulations)
- uncertainty in the initial carbon stock (based on observational constraints)

Since each component is based on the state-of-the-art science, our quantification of the uncertainties in the PCF is robust.

Reviewer #5 (Remarks to the Author):

In the manuscript entitled, "Climate policy implications of nonlinear decline of Arctic land permafrost and sea ice," the authors use dynamic emulators of complex physical models in the integrated assessment model PAGE-ICE to estimate the impacts of the nonlinear feedbacks on the global climate and economy under a range of climate scenarios consistent with the Paris Agreement. Largely, the revisions resolved the previous comments. However, there are a few lingering concerns.

The authors focus on two feedbacks in the Earth's climate system, the permafrost carbon feedback (PCF) and the sea-ice albedo feedback (SIAF), which could accelerate global warming and cause additional economic impacts. As these feedbacks represent three of the thirteen "tipping elements" identified in another recent study, the implication made by the authors is that actual impacts could be substantially more severe. However, these two feedbacks do not represent a complete picture of the feedbacks that constrain equilibrium climate sensitivity. More importantly, since radiative feedbacks (water vapor, clouds, sea ice, etc.) interact with each other, it is not well posed to suppose that the SIAF is the only one acting nonlinearly, or that nonlinear behavior in the SIAF will not affect the other feedbacks. For instance, Caldwell

et al. (2016) show significant covariances between albedo and shortwave cloud feedbacks, the latter which are negative in the Arctic. The consideration of the state-dependency of only a subset of high-latitude climate feedbacks is a limitation of the study and places a large caveat on the findings in regards to economic impacts.

We thank the Reviewer for these comments and acknowledge that we underappreciated the localised interactions between the surface albedo feedback (SAF) in the Arctic and other planetary feedbacks. As mentioned by the reviewer, this effect is most pronounced in the strong coupling between the SAF and shortwave cloud feedback in high latitudes, as demonstrated by Caldwell et al. (2016) in CMIP5 2xCO₂ equilibrium climate sensitivity (ECS) experiments.

The Caldwell et al. results are not applicable to transient climate simulations in our study. Therefore, to address the critical issue of the surface albedo & shortwave cloud coupling, we employed the four-parameter method of calculating the SAF using atmospheric transmissivity and reflectivity inferred from climate models by Winton (2005, 2006). The method involves a dynamic cloud parameterisation based on diagnostic variables from climate models, which allowed us to account for the changes in the Arctic cloud cover (and all other feedbacks represented by climate models) in response to variations in the surface albedo. Tests using CMIP5 models showed that the coupled cloud response in the Winton's method does reduce the strength of the Arctic SAF in transient simulations, which is similar to the negative covariances between the surface albedo and the clouds in the equilibrium analysis by Caldwell et al. The cloud parameterisation in the Winton's method also addresses the issue of high variability in the cloud feedback between climate models, which other authors dealt with by employing a fixed cloud approximation while calculating the SAF directly from the models' radiative kernels (Schneider et al., 2018).

To further address the Reviewer's concern regarding the limitation associated with considering only high latitude feedbacks, we used the Winton's method to calculate transient changes in the global SAF based on CMIP5 models. As expected from previous studies (Winton, 2006; Flanner, 2011), the resulting global SAF is dominated by the Arctic sea ice and land snow components, which at present account for around 75% of the global total SAF. The Arctic sea ice and land snow components are also responsible for the bulk of the decline in the global SAF with rising global mean surface temperature (GMST) (see Figures 3 and 6 in the revised manuscript). This cryosphere-driven decline in the global SAF has obvious implications for the climate policy results obtained using integrated assessment models (IAMs) that have so far relied on an over-estimated constant SAF as part of the 2xCO₂ ECS parameter. In fact, the persistent decline of the global SAF relative its average value implicitly included in the ECS parameter nearly offsets the additional warming from the permafrost carbon feedback (PCF) under the highest emission scenarios (see Figure 4 in the revised manuscript).

The other major global feedbacks such as clouds, water vapour and lapse rate contribute to the overall state-dependency in the ECS parameter. Their magnitudes have so far shown either weak responses to GMST or increases with GMST in climate model simulations (Colman and McAvaney, 2009; Meraner et al., 2013). While the state-dependencies in the planetary feedbacks require further investigations as part of CMIP6, the evidence so far suggests that apart from the SAF effects presented here, the magnitudes of the feedbacks are less likely to decrease with GMST. This implies that our estimates for the impacts of the state-dependent PCF and SAF are likely to be on a conservative side.

Because of all these considerations, we deemed it appropriate to keep the focus of the paper on the SAF and PCF, the two global feedbacks that appear to be highly nonlinear and misrepresented in climate policy studies. We further emphasise the role of the Arctic sea ice and land snow components of the global SAF, therefore maintaining the overall narrative of the importance of the nonlinear Arctic feedbacks for climate policy assessments. By explicitly modelling the land snow component, we also addressed another issue raised by the Reviewer (below).

As a minor aside, I was surprised by the authors' comment in their rebuttal that the snow albedo feedback is constant. Physically, I would expect it to operate under the same principle as the SIAF, i.e., the feedback approaches zero once no further melting can occur (as in Jonko et al., 2013). In the additional analyses the authors refer to, did they look at snow extent and/or snow albedo model output directly? I did not find the temperature amplification factor across the permafrost region to be a convincing proxy.

To address this issue, we used the Winton's method to explicitly represent the land snow component of the SAF, in addition to the sea ice component, as described above. We also modified the title of the manuscript accordingly.

However, separating the SAF into the Arctic sea ice, land snow and "rest of the world" components was required only for the diagnostic purposes, in order to show that the decline in the Arctic sea ice and land snow components are responsible for the bulk of the changes in the global SAF. In the revised manuscript, the results both for the GMST changes and for the additional economic impacts are presented for the nonlinear global SAF as a whole. Please see our responses to the previous comment by the same Reviewer for further details.

My second main comment pertains to the selection of CMIP5 models, and the meaning of uncertainty. The PCF emulator is driven by output from a different set of models than the SIAF emulator. The model mismatch compromises the physical consistency in the climate responses

used as input to the IAM. I don't see how this would provide an objective, physically based measure of uncertainty in climate or economic impacts. I am also troubled by the heavy constraints used as model selection criteria for the SIAF. Firstly, the models are required to match satellite-derived sea ice concentration and secondly to match the ensemble-mean projected sea ice extent. It is not obvious that fidelity to a short and imperfect observational record is sufficient to capture correctly the climate response to anthropogenic forcing (e.g., Sanderson and Knutti 2012, Boe and Terray 2015). In addition, this is a very different definition of uncertainty than the one used for the PCF, which was to "provide a plausible range of climate projections consistent with that of the entire CMIP5 models' set." I think the results are potentially misleading, especially where the effects of both feedbacks are combined, given the model selection process.

Once again, we thank the Reviewer for highlighting these critical issues, which have been underappreciated in the previous version of our manuscript. In order to address them, while also responding to other major points of concern raised both by this Reviewer and the other Reviewers, we adopted the following strategy in selecting and weighting the climate models and land surface models (LSM) (the latter are used to simulate permafrost carbon emissions):

- First, we have adopted the standard practice of using unweighted multi-model projections in our study, both for the PCF and SAF. We are aware and supportive of the ongoing efforts to define suitable model weighting criteria when combining the results of, for example, GCM simulations (Knutti et al., 2017). However, as this is a relatively new research area, we reverted to model democracy so as not to get side-tracked into debates of appropriate weighting strategies;
- Responding to Reviewers #3 and #4, we chose the SiBCASA and JULES LSMs to estimate the plausible range of permafrost responses to warming arising from uncertainty in LSM parameterisations;
- To capture the uncertainty in the PCF associated with climate forcing, both LSMs were forced with output from a range of climate models, sampling the full range of expected Arctic responses under a given climate scenario. SiBCASA was run to 2300 with output from five CMIP5 ESMs under two scenarios, whereas JULES was configured to run to 2300 with output from 22 CMIP3 ESMs under three climate scenarios;
- For the SAF, we used historic and RCP8.5 simulations of 16 CMIP5 ESMs that have the diagnostic variables required for the Winton's method. While short of the complete CMIP5 ensemble, these models sample the full range of Arctic responses as seen in the whole ensemble; 8 of the models have simulations extended out to 2300, which is important to capture the non-linear transitions for the SAF;
- While calculating the SAF using CMIP5 climate models, we did not apply bias-correcting to the sea ice, land snow and GMST simulations to preserve the internal consistency in the physics for each model.

As in the previous versions of the paper, we used the outputs from the climate and land system models to design and calibrate the emulators of the nonlinear PCF and SAF, which are statistical surrogates implemented within the PAGE-ICE IAM that represent the complex physical models. As PAGE-ICE is configured to run Monte-Carlo simulations, this approach allowed us to reproduce the uncertainties in the PCF and SAF and explore their climatic and economic impacts under a wide range of scenarios.

The new experimental design, including the model selection process, ensures that the uncertainties in the PCF and SAF are represented as well as is practically possible, while also maintaining the consistency of the approach throughout the paper.

Reviewer #2 (Remarks to the Author):

I am fully satisfied with the responses given to the 10 queries i raised and I also appreciate the effort done by the authors with the new analysis (zero emission scenario). As I said in my first referee report I think this is timely paper that contributes to a relevant line of research. Therefore, I recommend its publication in its current form.

Yours faithfully

The reviewer

We would like to thank the Reviewer once again for providing the useful comments that helped improve the manuscript.

References:

Caldwell, P.M., M.D. Zelinka, K.E. Taylor, and K. Marvel, 2016: Quantifying the Sources of Intermodel Spread in Equilibrium Climate Sensitivity. *J. Climate*, 29, 513–524, <https://doi.org/10.1175/JCLI-D-15-0352.1>

Jonko, A.K., K.M. Shell, B.M. Sanderson, and G. Danabasoglu, 2013: Climate Feedbacks in CCSM3 under Changing CO₂ Forcing. Part II: Variation of Climate Feedbacks and Sensitivity with Forcing. *J. Climate*, 26, 2784–2795, <https://doi.org/10.1175/JCLI-D-12-00479.1>

Boé, J. & Terray, L. *Clim Dyn* (2015) 45: 1913. <https://doi.org/10.1007/s00382-014-2445-5>

Sanderson, B. M., and R. Knutti (2012), On the interpretation of constrained climate model ensembles, *Geophys. Res. Lett.*, 39, L16708, doi: 10.1029/2012GL052665.

Burke, E. J., Ekici, A., Huang, Y., Chadburn, S. E., Huntingford, C., Ciais, P., ... & Krinner, G. (2017). Quantifying uncertainties of permafrost carbon–climate feedbacks. *Biogeosciences*, 14(12), 3051.

Schneider, A., Flanner, M., & Perket, J. (2018). Multidecadal Variability in Surface Albedo Feedback Across CMIP5 Models. *Geophysical Research Letters*, 45(4), 1972-1980.

Colman, R., & McAvaney, B. (2009). Climate feedbacks under a very broad range of forcing. *Geophysical Research Letters*, 36(1).

Meraner, K., Mauritsen, T., & Voigt, A. (2013). Robust increase in equilibrium climate sensitivity under global warming. *Geophysical Research Letters*, 40(22), 5944-5948.

Knutti, R., Sedláček, J., Sanderson, B. M., Lorenz, R., Fischer, E. M., & Eyring, V. (2017). A climate model projection weighting scheme accounting for performance and interdependence. *Geophysical Research Letters*, 44(4), 1909-1918

Reviewers' comments:

Reviewer #4 (Remarks to the Author):

The authors accounted for my two main comments (include other LSM, consider uncertainty in initial C stocks) in a satisfactory way. Regarding my points, I can recommend publication of the manuscript.

Reviewer #5 (Remarks to the Author):

Review for manuscript entitled "Climate policy implications of nonlinear decline of Arctic land permafrost, snow and sea ice" by Dr Yumashev and colleagues (manuscript number NCOMMS-17-30910B).

The authors have addressed my previous comments, and for the most part I'm satisfied with the revisions. In particular, I think the use of multiple LSMs, evaluation of the land snow component of the surface albedo feedback, and implementation of climate model democracy have all improved the manuscript. I have two remaining comments, and one very minor one.

First, I don't think the revision is accounting for the cloud feedback as the authors claim at the bottom of p. 4 of the manuscript. "The method involves a dynamic cloud parameterisation based on diagnostic variables from GCMs, which allows us to account for the changes in the Arctic cloud cover *in response to variations in the surface albedo* ..." (emphasis mine). I don't see how the Winton method, using diagnostic variables only, would capture the cloud *response* to surface albedo changes. I wonder if the authors are referring to a correction to the feedback estimate to remove the effect of cloud changes on surface shortwave fluxes, and thus to isolate the portion only due to surface albedo changes. This would seem analogous to cloud masking, which is incorporated in kernel based estimates of cloud feedbacks (e.g., Soden et al. 2008), to account for the fact that the cloud radiative effect (difference in all sky and clear sky fluxes) will change if surface albedo changes, even for no change in cloudiness. To be clear, I don't have an issue with the SAF analysis in the revised manuscript, but I am concerned it's being over-interpreted in this one respect. As far as I can tell, the authors have not calculated a shortwave cloud feedback, so I don't see how their calculation "addresses the issue of high variability in the cloud feedback between climate models." My recommendation would simply be to temper or revise some of these statements. Later, in the section of Robustness, the authors refer to the role of other feedbacks, and that is probably sufficient for this study.

Second, I am concerned about the cases of no statistical significance being buried in the footnote. "However, the values plotted in Figure 4 represent statistically significant shifts in the state of the climate system due to the two feedbacks at the 95% confidence level." The footnote then goes on to contradict this statement. That is, how can Figure 4 represent statistically significant shifts if, for instance, PCF & SAF combined from the second half of the 22nd century onwards is not significant?

Minor comment in regard to Figure 5: The long tails for BaU and NDC-P are off the plot. The values for the upper end of the 5-95% range should be included somewhere—in the text if not in the figure itself.

NCOMMS-17-30910B “Climate policy implications of nonlinear decline of Arctic land permafrost, snow and sea ice”

Dmitry Yumashev et al.

Response to Referees

We are grateful to the Reviewers for the additional comments on our manuscript, and welcome the opportunity to address the outstanding issues in the new revision presented here.

Generic updates

To comply with the general formatting requirements for Nature Communications, the revised manuscript includes the following updates:

- The Abstract has been shortened to 150 words and meets the key presentation requirements
- The last paragraph in the Introduction now outlines the key results and conclusions of the paper. There are also no display items in the Introduction
- The paragraph dedicated to the finding that the 1.5 and 2°C scenarios are statistically equivalent, including a number of arguments why it would be prudent to aim for emissions towards the lower end of this temperature range, has been moved from the Results to the Discussion. There are no overlaps between the two sections

Additional clarifications

We added another figure to illustrate the statistical equivalence of the 1.5 and 2°C scenarios (Figure 6). The key statistics of the probability distribution provided in the legend to the figure also shows that considering the nonlinear Arctic feedbacks makes the 1.5°C target marginally more economically attractive than the 2°C target. As these are some of the main results presented in the paper, we deem the extra illustration necessary. This brings the total for the display items to 9 (8 figures, 1 table).

We also included a brief description of the small peaks in the sea ice and land snow SAF components in the range of GMST anomalies between 0 and 3°C, which are clearly visible in Figure 2 (Results section). The peaks coincide with seasonal changes in the summer sea ice (Stroeve et al., 2012; Sigmond et al., 2018) and spring and summer land snow (Shi and Wang, 2015) covers, coupled with high Arctic insolation. In addition, the plateau in the sea ice SAF component between 5 and 7°C coincides with the loss of the spring sea ice (Winton, 2006b;

Hezel et al., 2014). However, establishing the exact causes of the nonlinear transitions in the individual SAF components is less important given the nearly monotonic decline seen in the global SAF (Figure 8 of the paper), which is at the basis of the reported results.

Numerical updates

We made minor corrections to the SAF calibration for the high-end of the simulated GMST range (above 10°C), resulting in the effect of the nonlinear SAF becoming slightly more negative for the high emission scenarios. The text and the relevant plots have been updated accordingly.

Reviewers' comments:

Reviewer #4 (Remarks to the Author):

The authors accounted for my two main comments (include other LSM, consider uncertainty in initial C stocks) in a satisfactory way. Regarding my points, I can recommend publication of the manuscript.

We would like to thank the Reviewer once again for providing the useful comments that helped improve the manuscript.

Reviewer #5 (Remarks to the Author):

Review for manuscript entitled "Climate policy implications of nonlinear decline of Arctic land permafrost, snow and sea ice" by Dr Yumashev and colleagues (manuscript number NCOMMS-17-30910B).

The authors have addressed my previous comments, and for the most part I'm satisfied with the revisions. In particular, I think the use of multiple LSMs, evaluation of the land snow component of the surface albedo feedback, and implementation of climate model democracy have all improved the manuscript. I have two remaining comments, and one very minor one.

*First, I don't think the revision is accounting for the cloud feedback as the authors claim at the bottom of p. 4 of the manuscript. "The method involves a dynamic cloud parameterisation based on diagnostic variables from GCMs, which allows us to account for the changes in the Arctic cloud cover **in response to variations in the surface albedo** ..." (emphasis mine). I don't see how the Winton method, using diagnostic variables only, would capture the cloud response to surface albedo changes. I wonder if the authors are referring to a correction to*

the feedback estimate to remove the effect of cloud changes on surface shortwave fluxes, and thus to isolate the portion only due to surface albedo changes. This would seem analogous to cloud masking, which is incorporated in kernel based estimates of cloud feedbacks (e.g., Soden et al. 2008), to account for the fact that the cloud radiative effect (difference in all sky and clear sky fluxes) will change if surface albedo changes, even for no change in cloudiness. To be clear, I don't have an issue with the SAF analysis in the revised manuscript, but I am concerned it's being over-interpreted in this one respect. As far as I can tell, the authors have not calculated a shortwave cloud feedback, so I don't see how their calculation "addresses the issue of high variability in the cloud feedback between climate models." My recommendation would simply be to temper or revise some of these statements. Later, in the section of Robustness, the authors refer to the role of other feedbacks, and that is probably sufficient for this study.

We thank the Reviewer for pointing this issue out and agree that it requires clarification and amendments to the text.

Winton's ALL/CLR method to compute the SAF uses a parameterization for upward atmospheric reflectivity (Winton 2006, Eq. 4). It depends on the relation between downward shortwave fluxes at the surface under all skies (with clouds) and clear skies (assumed to be no clouds). According to Winton (2006a), the "reflectivity is mainly due to reflections from the undersides of clouds and will depend upon cloud parameters such as cloud water path and effective drop radius", while "the coefficients in Eq. (4) are round numbers based on physical reasoning and are not fit to any particular model result (Winton 2005)." Therefore, Winton's ALL/CLR method includes the **effect of clouds** on the value of the SAF. However, as the reviewer correctly pointed out, the method does not distinguish between changes in the cloud cover (cloudiness) in warmer climates (Collins et al., 2013) and the localised changes in the clouds driven specifically by changes in the surface albedo, which were isolated using radiative kernels (Soden et al., 2008). The reviewer is also correct that we do not compute the **cloud feedback**, which is included implicitly in the ECS parameter in the PAGE-ICE model and is assumed to be state-independent (please see the Robustness section). We revised the text accordingly, as shown below.

The effect of clouds on the SAF in the Winton's method can be demonstrated in a simple experiment. Winton (2006a) computes the change in shortwave radiation flux (W/m²) as the difference between a reference period and a perturbation experiment (Eq. 6). This difference depends on both the reference and perturbed values of the upward and downward shortwave radiation, surface albedo and atmospheric reflectivity. As mentioned above, the atmospheric reflectivity includes the effect of clouds. We conducted a simple experiment where the atmospheric reflectivity always represents the reference period (which in our case is the 1850-1900 baseline pre-industrial climatology) and thus the cloudiness remains constant, while the other variables change during the perturbation experiment (30-year

climatological windows moving from the reference period until 2100 or 2300 under RCP8.5). We refer to this as the “fixed pre-industrial cloud” experiment. The effect of time-constant cloud cover fixed at the pre-industrial climatology is to increase the SAF for all of its three main components considered in the study (sea ice, land snow and rest of the world), which is illustrated in the Figure below. This result could be interpreted as follows. An increased global cloudiness for transient climate simulations under RCP8.5 (e.g., Collins et al., 2013) reflects more shortwave radiation back into the atmosphere compared with the cloud cover from the pre-industrial period, which leads to a small reduction in the actual SAF relative to its hypothetical value computed with the fixed pre-industrial cloud cover.

As mentioned earlier, the fixed pre-industrial cloud experiment cannot tell us whether the dampening effect of the clouds on the sea ice and land snow SAF components is associated specifically with the localised changes to the surface albedo (e.g., Soden et al., 2008; Caldwell et al., 2016), or whether it is mainly driven by the overall global changes to cloudiness in warmer climates. Since clouds are part of the fully coupled CMIP5 climate models, both of these effects are implicitly included in the diagnostic shortwave variables used in the ALL/CLR method of calculating the SAF (Winton, 2005; 2006a) that we adopted here.

Figure. SAF components for Arctic sea ice (a), land snow (b) and rest of the world (c) as functions of the GMST rise relative to the 1850-1900 pre-industrial conditions. Obtained from multiple CMIP5 GCMs using Winton’s ALL/CLR method, assuming dynamic (red) and fixed pre-industrial (blue) cloud covers. Lines: multi-model mean; shaded areas: ± 1 SD.

Revised text

The original text in question read:

“We base the SAF emulator on the four-parameter method of calculating the SAF using downward and upward atmospheric transmissivity and reflectivity inferred from climate models (GCMs) (Winton, 2005; 2006a). The method involves a dynamic cloud parameterisation based on diagnostic variables from GCMs, which allows us to account for the changes in the Arctic cloud cover in response to variations in the surface albedo, the effect known to reduce the contribution of the high-latitude SAF to the 2xCO₂ ECS parameter (Caldwell et al., 2016). The cloud parameterisation in the SAF calculation also addresses the issue of high variability in the cloud feedback between climate models (Schneider et al., 2018).”

To address the comments by the reviewer and reflect on the clarifications made above, we changed this paragraph to the following:

“We base the SAF emulator on the ALL/CLR method of calculating the SAF using downward and upward atmospheric transmissivity and reflectivity inferred from climate models (GCMs) (Winton, 2005; 2006a). The method involves an atmospheric reflectivity parameterisation, which represents the effect of clouds and is based on clear sky and all sky shortwave fluxes diagnosed from the GCMs. It allows us to account for the localised changes to the cloud cover and its effect on the SAF in line with the physical interactions represented in the fully-coupled CMIP5 models (Soden et al., 2008; Caldwell et al., 2016; Schneider et al., 2018). We do not compute the global cloud feedback; instead, it is included implicitly in the ECS parameter in the PAGE-ICE model and is assumed to be state-independent (footnote: see the section on the robustness of the results below).”

Second, I am concerned about the cases of no statistical significance being buried in the footnote. “However, the values plotted in Figure 4 represent statistically significant shifts in the state of the climate system due to the two feedbacks at the 95% confidence level.” The footnote then goes on to contradict this statement. That is, how can Figure 4 represent statistically significant shifts if, for instance, PCF & SAF combined from the second half of the 22nd century onwards is not significant?

We acknowledge that in its current form the statement appears to contradict the footnote, therefore making the cases of no statistical significance less apparent. To address this issue, we modified the sentence in question to explicitly state that there are a few exceptions when the shift to the climate system is not significant at the 95% confidence level; the exceptions themselves are now listed in the caption to Figure 4. The revised sentence therefore reads:

“However, with few exceptions, the values plotted in Figure 4 represent statistically significant shifts in the state of the climate system due to the two feedbacks at the 95% confidence level (the exceptions are listed in the Figure 4 caption).”

We believe this makes the exceptions more transparent and accessible to the reader.

Minor comment in regard to Figure 5: The long tails for BaU and NDC-P are off the plot. The values for the upper end of the 5-95% range should be included somewhere—in the text if not in the figure itself.

We increased the range of the y-axis on the plots to include the previously omitted upper ends of the 5-95% ranges for BaU and NDC-P.

References:

Caldwell, P. M., Zelinka, M. D., Taylor, K. E., & Marvel, K. (2016). Quantifying the sources of intermodel spread in equilibrium climate sensitivity. *Journal of Climate*, 29(2), 513-524.

Collins, M., R. Knutti, J. Arblaster, J.-L. Dufresne, T. Fichet, P. Friedlingstein, X. Gao, W.J. Gutowski, T. Johns, G. Krinner, M. Shongwe, C. Tebaldi, A.J. Weaver and M. Wehner, 2013: Long-term Climate Change: Projections, Commitments and Irreversibility, in: *Climate Change 2013: The Physical Science Basis. Contribution of Working Group I to the Fifth Assessment Report of the Intergovernmental Panel on Climate Change* [Stocker, T.F., D. Qin, G.-K. Plattner, M. Tignor, S.K. Allen, J. Boschung, A. Nauels, Y. Xia, V. Bex and P.M. Midgley (eds.)], Cambridge University Press, Cambridge, UK and New York, NY, USA, p1070

Hezel, P. J., Fichet, T., & Massonnet, F. (2014). Modeled Arctic sea ice evolution through 2300 in CMIP5 extended RCPs. *The Cryosphere*, 8(4), 1195-1204.

Schneider, A., Flanner, M., & Perket, J. (2018). Multidecadal Variability in Surface Albedo Feedback Across CMIP5 Models. *Geophysical Research Letters*, 45(4), 1972-1980.

Shi, H. X., & Wang, C. H. (2015). Projected 21st century changes in snow water equivalent over Northern Hemisphere landmasses from the CMIP5 model ensemble. *The Cryosphere*, 9(5), 1943-1953.

Sigmond, M., Fyfe, J. C., & Swart, N. C. (2018). Ice-free Arctic projections under the Paris Agreement. *Nature Climate Change*, 8(5), 404.

Soden, B. J., Held, I. M., Colman, R., Shell, K. M., Kiehl, J. T., & Shields, C. A. (2008). Quantifying climate feedbacks using radiative kernels. *Journal of Climate*, 21(14), 3504-3520.

Stroeve, J. C., Serreze, M. C., Holland, M. M., Kay, J. E., Malanik, J., & Barrett, A. P. (2012). The Arctic's rapidly shrinking sea ice cover: a research synthesis. *Climatic Change*, 110(3), 1005-1027.

Winton, M. (2005). Simple optical models for diagnosing surface-atmosphere shortwave interactions. *Journal of Climate*, 18(18), 3796-3805.

Winton, M. (2006a). Surface albedo feedback estimates for the AR4 climate models. *Journal of Climate*, 19(3), 359-365.

Winton, M. (2006b). Does the Arctic sea ice have a tipping point? *Geophysical Research Letters*, 33(23).

REVIEWERS' COMMENTS:

Reviewer #5 (Remarks to the Author):

The authors have addressed my previous comments satisfactorily. I recommend publication of the manuscript.